# Characterization of Cme and Yme thermostable Cas12a orthologs

Ryan T. Fuchs [1], Jennifer L. Curcuru [1], Megumu Mabuchi[1], Audrey Noireterre [1,2], Peter R. Weigele [1], Zhiyi Sun [1] & G. Brett Robb [1✉]

CRISPR-Cas12a proteins are RNA-guided endonucleases that cleave invading DNA containing target sequences adjacent to protospacer adjacent motifs (PAM). Cas12a orthologs have been repurposed for genome editing in non-native organisms by reprogramming them with guide RNAs to target specific sites in genomic DNA. After single-turnover dsDNA target cleavage, multiple-turnover, non-specific single-stranded DNA cleavage in *trans* is activated. This property has been utilized to develop in vitro assays to detect the presence of specific DNA target sequences. Most applications of Cas12a use one of three well-studied enzymes. Here, we characterize the in vitro activity of two previously unknown Cas12a orthologs. These enzymes are active at higher temperatures than widely used orthologs and have subtle differences in PAM preference, on-target cleavage, and *trans* nuclease activity. Together, our results enable refinement of Cas12a-based in vitro assays especially when elevated temperature is desirable.

[1] New England Biolabs Inc, Ipswich, MA 01938, USA. [2] Département de Biologie Cellulaire (BICEL), Université de Genève, CH - 1211 Genève 4, Switzerland.
✉email: robb@neb.com

Clustered Regularly Interspaced Short Palindromic Repeats (CRISPR) are DNA sequences associated with acquired immune response systems in bacteria and archaea[1–3]. There is extensive diversity amongst the known CRISPR systems, but a common characteristic is that they use an RNA guide and effector protein or protein complex to form a ribonucleoprotein (RNP) that targets and cleaves foreign nucleic acids[4,5]. The effector proteins Cas9 and Cas12a (previously known as Cpf1) from Class 2 Type II and Type V-A CRISPR systems, respectively, use RNA guides to cleave dsDNA targets and have been repurposed for genome editing applications in eukaryotic and prokaryotic organisms[6–16].

Cas12a requires a ~40 nt guide RNA (crRNA). The 3′-terminal ~20 nt of the crRNA imparts sequence specificity to a dsDNA target bearing a T-rich protospacer adjacent motif (PAM) adjacent to the target sequence[11]. After pairing the crRNA sequence to the target DNA sequence, Cas12a uses a single RuvC domain to sequentially cleave each strand of the DNA target to generate 5′ overhangs and the Cas12a RNP remains bound to the PAM-proximal cleaved DNA while the PAM-distal DNA dissociates[11,17–22]. Recent work has identified an additional activity attributed to Cas12a, specifically that the RNP remains in an activated state after DNA cleavage while it remains bound to the PAM-proximal DNA. The activated state leads to collateral, non-specific degradation of ssDNA in trans (trans nuclease activity) that is attributed to the RuvC domain[23–27]. The activated RuvC carries out multiple-turnover cleavage of non-target ssDNAs in trans, likely enabled by its solvent accessibility after dissociation of the PAM-distal fragment of cleaved target DNA[25]. This property of Cas12a has been utilized to develop programmable sequence-specific DNA detection and diagnostic tests where target nucleic acids are amplified using isothermal amplification approaches[23,24,26,28]. These assay platforms called DETECTR[23], SHERLOCKv2[26], and HOLMES[24] utilize Cas12a to target a DNA sequence of interest. Upon cleavage of the sequence of interest, the trans nuclease activity degrades short ssDNA oligos present in the reaction that have a fluorophore on one terminus and a quencher on the other resulting in an increase in fluorescence over time. These approaches have been proven to be a fast and sensitive way to detect viral nucleic acid during the current SARS-CoV-2 pandemic[28–38]. Isothermal amplification approaches including reverse transcription loop-mediated amplification (RT-LAMP) and helicase-dependent amplification (HDA) are typically performed at moderately high temperatures such as 65 °C[39,40]. Cas12a enzymes that are active at higher temperatures would be desirable to streamline coupled isothermal amplification and Cas12a trans nuclease-based detection.

In this work, we characterize two Cas12a orthologs Compost metagenome (Cme) and Yellowstone metagenome (Yme) that we identified in metagenomic sequence databases and compare their properties to those of the three most well-studied Cas12a orthologs, *Lachnospiraceae* bacterium ND2006 (Lba), *Francisella tularensis* subsp. *novicida* strain U112 (Fno) and *Acidaminococcus sp.* BV3L6 (Asp)[11]. We show that Cme and Yme are naturally thermotolerant proteins in that they have both RNA-guided, targeted dsDNA cleavage and trans nuclease activity at elevated temperatures at which wild-type Asp, Fno, and Lba-Cas12a orthologs are inactive. We also report differences between the enzymes in PAM sequence requirements for dsDNA cleavage, the ends of products of RNA-guided target dsDNA cleavage, substrate preferences for trans nuclease activity, and the ability to modulate trans nuclease activity by varying reaction conditions. Taken together, our results highlight the underappreciated biochemical diversity of Cas12a enzymes and their properties for in vitro applications. The thermotolerance of Cme and Yme-Cas12a has the potential to be beneficial for assays where it is desirable to detect DNA targets at elevated temperatures.

## Results

**Identification of Cme and YmeCas12a**. We scanned publicly available metagenome contigs for loci containing both predicted CRISPR-Cas12a proteins and CRISPR arrays and identified candidates of interest in sequences obtained from the Joint Genome Institute's Integrated Microbial Genomes & Metagenomes resource[41] (IMG/M): https://img.jgi.doe.gov/cgi-bin/m/main.cgi. We named two of these candidates **C**ompost **me**tagenome (Cme) Cas12a and **Y**ellowstone **me**tagenome (Yme) Cas12a, respectively. The Cme protein sequence is 1288 amino acids in length and shares 39.9, 31.5, and 45.2% amino-acid identity to the well-studied Lba, Asp, and FnoCas12a orthologs, respectively. Yme is 1362 amino acids in length and has <27% amino-acid identity to any of the other orthologs. We constructed a phylogenetic tree to visualize the relatedness of the orthologs at the primary amino-acid level using 100 Cas12a sequences from The National Center for Biotechnology Information (https://www.ncbi.nlm.nih.gov/) and various publications[11,42–44]. The tree shows that the five different orthologs studied here are representatives of unique Cas12a clusters (Fig. 1a), suggesting that they are evolutionarily distinct from each other. The DNA sequences corresponding to the crRNA sequences for both Cme and Yme were identifiable in the metagenomic loci and the 5′-stem-loop sequence and the structure characteristic of Cas12a crRNAs is highly homologous to other characterized Cas12a orthologs[11] (Fig. 1b).

**Cme and Yme are thermotolerant Cas12a orthologs**. Since the sequences for Cme and Yme were identified in metagenomic sources associated with warm temperatures, we wanted to assess the thermostability of these enzymes. First, we obtained RNA oligonucleotides corresponding to the sequences shown in Fig. 1b on the 5′-end and 20 nt of the target sequence on the 3′-end of the oligonucleotide. Next, we performed experiments to determine the point at which the Cas12a proteins lacking crRNA (apo) or with their corresponding crRNA (RNP) unfold using nanoDSF and a temperature range from 20 °C to 80 °C with a rate of change of 1 °C sec$^{-1}$. Prior to this, we confirmed the formation of Cas12a RNPs using size exclusion chromatography monitoring shifts in peak elution volume for apo protein versus RNP (Supplementary Figure S1). Raw traces of nanoDSF data can be found in Supplementary Figure S2. As shown in Fig. 2a, Cas12a orthologs in the apo form unfold in the low to mid 40 °C range for Lba, Asp, and Fno, but apo CmeCas12a is stable until ~54 °C and YmeCas12a until ~60 °C. Melting points of all Cas12a RNPs were higher than apo proteins and we observed that AspCas12a and FnoCas12a exhibited substantially increased melting temperatures of ~54 °C for AspCas12a, and ~55 °C for FnoCas12a. We measured the melting point of Cme and YmeCas12a RNPs to be ~56 °C and ~65 °C, respectively, higher than the other orthologs tested. These results indicate that Cme and YmeCas12a are thermostable as apo proteins and RNPs.

We compared in vitro dsDNA cleavage activity of Cme and YmeCas12a to Lba, Asp, and FnoCas12a at various temperatures. As shown in Fig. 2, all five Cas12a orthologs cleave a large percentage of the substrate DNA at temperatures below their unfolding point. We observed a broad active temperature range for Lba, Asp, and FnoCas12a (Fig. 2b) when RNPs were formed at 25 °C. However, when Lba, Asp, and FnoCas12a RNPs were formed at reaction temperature we failed to detect cleavage activity above 45 °C (Fig. 2c). Cme and YmeCas12a retain dsDNA cleavage activity at higher temperatures than Lba, Asp, and FnoCas12a when RNPs were formed at 25 °C or when RNPs were formed at higher temperatures. This difference was especially noticeable in the latter case. Thus, Cme and YmeCas12a are

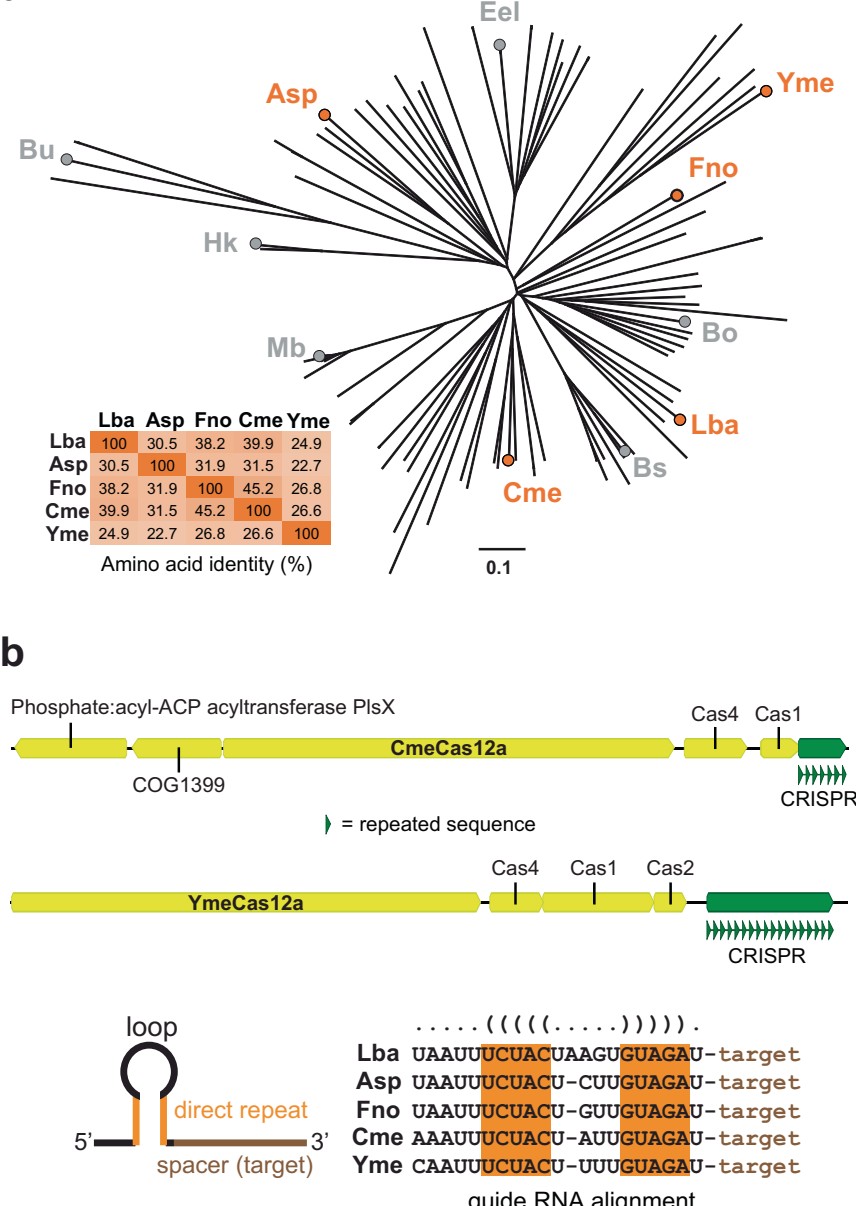

**Fig. 1 Relatedness of Cas12a orthologs. a** Phylogenetic tree of 100 Cas12a protein sequences. The branches of the tree corresponding to the five proteins examined in this study are marked with an orange circle and the table shows their percentage of shared amino-acid sequence. **b** Cartoon representation of the Cme and Yme loci from metagenomic data and an alignment of Cas12a crRNA sequences. DNA CRISPR repeat sequences shown in green are representative of the crRNA sequence utilized by the corresponding Cas12a protein. A typical crRNA is also represented as a cartoon and has 20 nt of target sequence appended to the 3′-end of the sequence shown in the alignment next to cartoon.

active, thermostable nucleases with robust, targeted dsDNA cleavage activity.

**PAM preference and tolerance of sub-optimal PAMs**. The TTTA sequence included in substrates in Fig. 2, is a canonical PAM recognized by Lba, Asp, and FnoCas12a[11]. To our knowledge, all characterized Cas12a orthologs recognize a pyrimidine-rich PAM at positions −2, −3, and usually −4 nt relative to the target sequence[11,44–47]. Although we predicted that this was likely to also be the case with Cme and YmeCas12a, we performed detailed comparisons of PAM sequences required for on-target dsDNA cleavage. To that end, we utilized an assay in which the dsDNA substrates are 124 bp dsDNA minicircles with 10 nt of randomized PAM sequence adjacent to a target sequence[48]

(Supplementary Figure S3). Libraries of circular dsDNA substrates were exposed to Cas12a RNPs loaded with crRNA corresponding to the target and incubated for 2 min at 37 °C for Lba, Asp, and Fno and 55 °C for Cme and Yme. Linearized DNA was end-repaired, ligated to adapters, and subjected to Illumina sequencing. The remaining circular DNA could not be sequenced since it lacked free ends for adapter ligation. Figure 3 shows, in graphical form, position weight values calculated by using the frequency of nucleotides at each of the 10 randomized PAM positions in the Cas12a RNP exposed samples normalized to a BstXI restriction enzyme digested sample as described in the methods. The BstXI digested sample serves as a control to determine the nucleotide frequency in the PAM positions prior to Cas12a exposure.

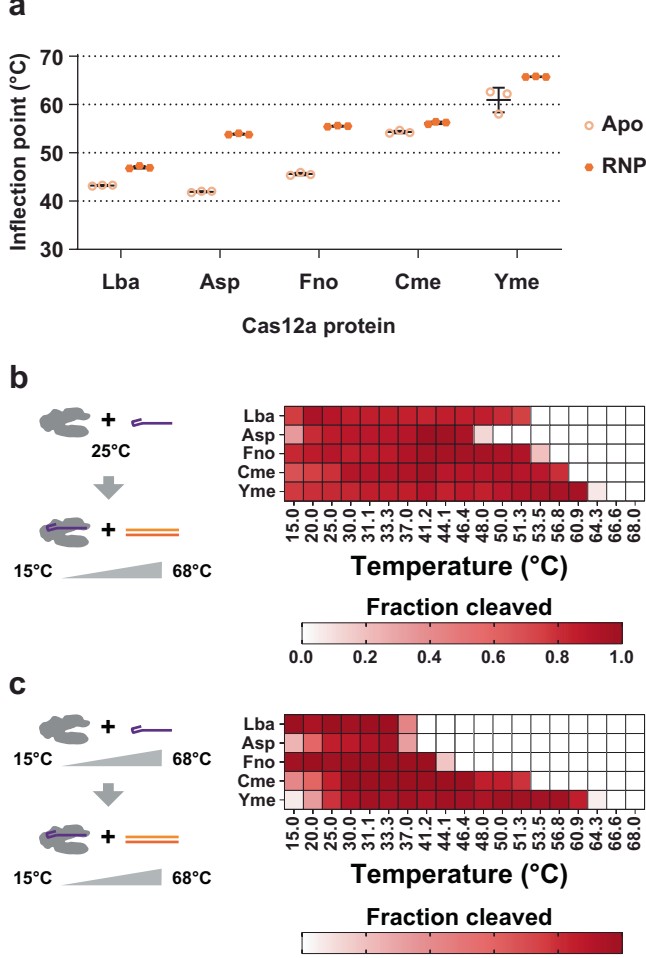

**Fig. 2 Cas12a thermostability and dsDNA target cleavage activity. a**
Thermal stability of Cas12a proteins without crRNA (apo) or loaded with crRNA (RNP). Protein unfolding was measured with nano differential scanning fluorometry (NanoDSF) over a temperature range from 20 °C to 80 °C. Fluorescence was monitored as temperature increased at a rate of 1 °C sec$^{-1}$. The inflection point of the fluorescent curve is interpreted as the unfolding point of the protein. Data points collected from replicate experiments are plotted and error bars represent mean ±standard deviation. Raw traces of the fluorescence data can be found in Supplementary Figure S2. The on-target double-stranded DNA (dsDNA) cleavage activities of Cas12 RNPs was measured using in vitro assays containing fluorophore-labeled dsDNA target substrates. Cas12a protein was incubated with crRNA at either 25 °C (**b**) or at varying temperatures (**c**) to form RNPs. RNPs formed at 25 °C were then transferred to the indicated reaction temperature. A cleavage reaction was initiated by adding 5′ fluorescein-labeled target DNA pre-incubated at the reaction temperature to the Cas12a RNPs. Cleaved fragments were quantified and the proportion of the substrate cleaved in each condition is represented in a heatmap where the intensity of the red color corresponds to the extent of target cleavage. Source data used to make the graphs are provided in Supplementary Data 2.

We identified TTTV PAM preferences for Lba, Asp, and FnoCas12a—similar to those reported in the literature[11]. We observed that the preference for CmeCas12a was also TTTV although the nucleotide preference at position −4 was noticeably less than that of positions −2 or −3. We observed a more pronounced reduction in nucleotide preference at the −4 position for YmeCas12a, which contrasts with the clear preference

observed at positions −2 and −3. In addition, we observed that "C" is not depleted as strongly as "A" or "G" at positions −4, −3, and −2 for Asp, Cme, and YmeCas12a, suggesting that it is tolerated in the PAM sequence as has been reported for the other Cas12a orthologs[42,49,50]. We observed similar PAM preferences using substrates with a different target-crRNA combination (Supplementary Figure S4).

To better understand PAM preference in vitro, we generated a panel of fluorescein-labeled DNA substrates with various defined PAM sequences at the −4, −3, and −2 positions relative to a target sequence. We used the panel for in vitro cleavage assays where the DNA substrates and a 10-fold mass excess of HeLa genomic DNA were used to mimic an amplified target region in a DNA sample. DNA substrates were incubated with Cas12a RNPs and aliquots were removed at 1, 5, and 10 min. For each time point and DNA substrate, we quantified the proportion of substrate cleaved. As shown in Fig. 4, all Cas12a orthologs showed the greatest extent of cleavage on substrates containing a "T" at PAM positions, −2 and −3, which fits the canonical Cas12a TTTV PAM sequence. Remarkably, we observed the cleavage of DNA substrates containing non-canonical PAMs. For example, all orthologs cleaved substrates containing a purine at position −4 if positions −2 and −3 were "T". This is consistent with the data in Fig. 3 where there appears to be a weaker preference at the −4 position for all orthologs except LbaCas12a. Notably, LbaCas12a produced the lowest activity on RTT PAMs of the four orthologs, consistent with its preference of T at position −4. All orthologs were able to tolerate other non-canonical PAMs; in fact, nearly all sequence combinations containing either 2 or 3 pyrimidines at positions −2, −3, or −4 were cleaved at least to some extent. One exception is that Asp was the only enzyme able to produce cleavage with a YRY PAM. Notably, we observed no cleavage with PAMs that contained 0 or only 1 pyrimidine residue. We repeated these experiments in reactions lacking HeLa genomic DNA as a decoy (Supplementary Figure S5). The results show a similar trend as those in Fig. 4 except the accumulation of cleaved products was much faster for all orthologs, presumably because they could find and cleave target DNA faster without excess HeLa DNA present, and there was a small amount of activity observed for PAMs with only 1 pyrimidine. Taken together, these results support previous findings that TTT at positions −4, −3, and −2 is the optimal PAM for Cas12a nucleases, but it is likely that most NYY sequences at those positions can be used as PAMs, at least to some extent, for in vitro applications.

**Cleavage profiles of Cas12a orthologs are diverse**. Cas12a nucleases cleave their dsDNA targets to yield staggered 5′ overhangs, but the cleavage site on each strand of the DNA is known to be imprecise[11,51,52]. We sought to characterize the dsDNA products resulting from Cme and YmeCas12a on-target cleavage. An advantage of the approach that we utilized for PAM identification is that once the small, dsDNA circular molecules are linearized, they are short enough to be sequenced from end to end in an Illumina sequencing run (Fig. 5a). In addition, the end repair step of preparing a library for sequencing encodes information about the nature of the double-stranded break generated by the linearization event in both strands of the linearized molecule. Thus, every single strand of library molecule contains information about the cleavage site for each double-stranded cleavage event.

We performed control reactions with restriction enzymes FspI, HhaI, or HinP1I that leave blunt, 3′ overhang, or 5′ overhangs, respectively. For all three restriction enzymes, we observed that >97% of the mapped reads corresponded to their expected cleavage site on both strands of DNA (Supplementary Figure S6).

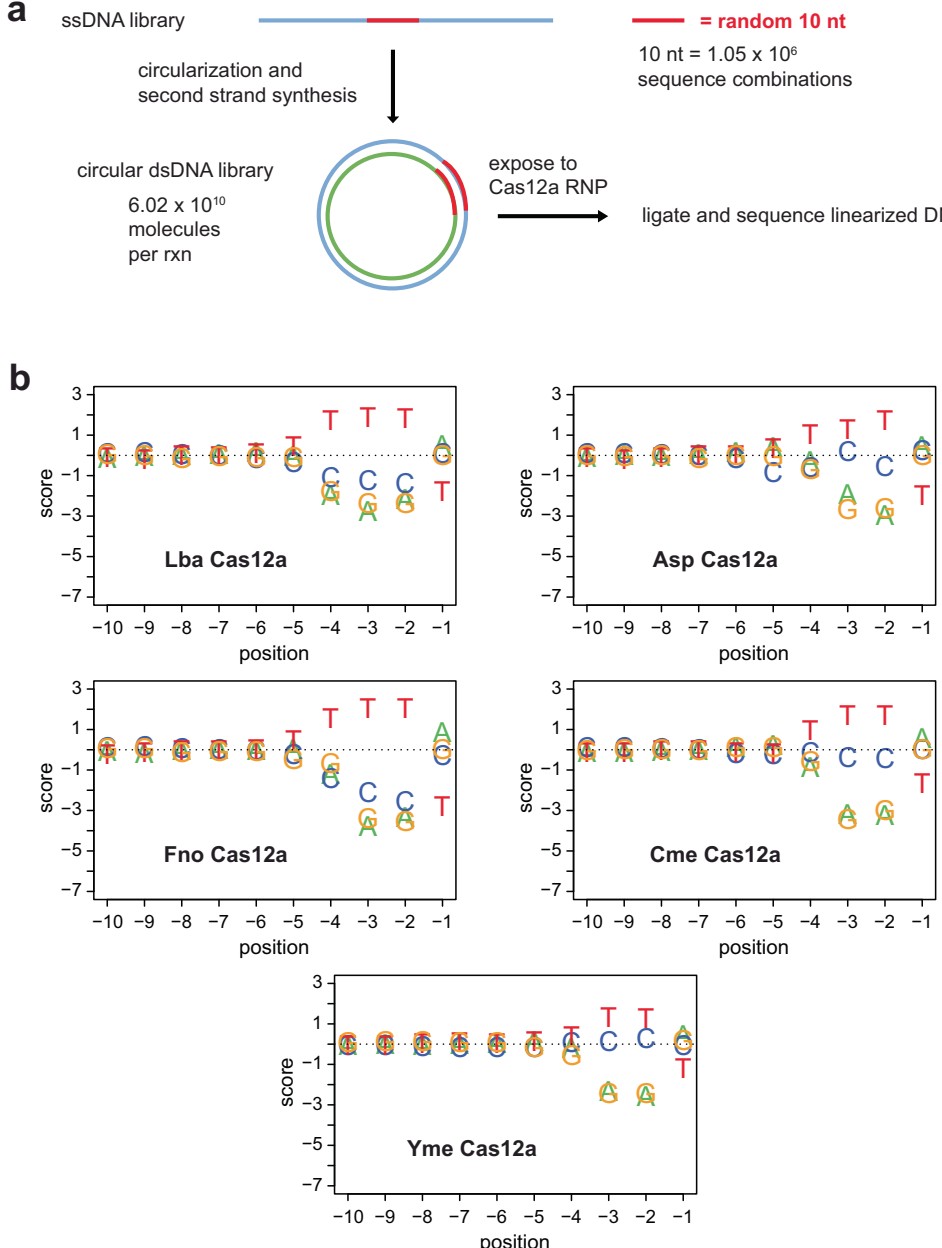

**Fig. 3 PAM requirements of Cas12a orthologs. a** Schematic representation of the workflow used to determine PAM sequence requirements for double-stranded DNA (dsDNA) cleavage. Single-stranded DNA oligonucleotide libraries with 10 nt of randomized sequence adjacent to a 20 nt target sequence were converted into dsDNA minicircle libraries and exposed to Cas12a RNPs. Linearized dsDNA libraries were subjected to end repair, adapter ligation, and sequencing. A detailed, step-wise diagram of this method is provided in Supplementary Figure S3. **b** PAM sequence requirements of Cas12a orthologs. Position weight values were calculated as described in the methods and are shown as graphs where a positive score corresponds to an enrichment of a particular base in the randomized 10 nt region on a $\log_2$ scale and a negative score corresponds to depletion. The position is the distance in nt relative to the first nt of the target sequence. Sequences of oligonucleotides used to make circular substrate DNAs can be found in Supplementary Data 11.

Next, we produced circular dsDNA substrates with TTTA PAMs flanking one of five different target sequences with GC content of 30%, 45%, 50%, 55%, and 70% (Supplementary Table S1). We subjected the substrates to digestion with Cas12a RNPs containing corresponding crRNA at 37 °C and observed that Lba, Asp, and FnoCas12a RNP digests accumulated DNA ends with 5′ overhangs, consistent with previous studies[11] and both Cme and YmeCas12a RNP digests accumulated 5′ overhangs as well (Fig. 5b; additional data in Supplementary Data 2). Although cleaved DNA ends were expected to be heterogeneous for Cas12a nucleases, we observed unexpected differences in the

pattern of cleaved DNA ends—both between orthologs and between targets. All five Cas12a orthologs almost exclusively accumulated cleaved DNA ends terminating at one or two adjacent bases on the target strand (TS), but accumulated termini of the non-target strands (NTS) were more diverse. This result is consistent with the observation that AspCas12a forms a gap in the NTS prior to TS cleavage[52]. Despite this, the diversity in NTS termini varies between targets with some targets accumulating many termini on the NTS while others accumulated only one or two major ends. Heterogeneity of ends on the NTS was highest for Asp, while Lba, Fno, and Cme were intermediate and

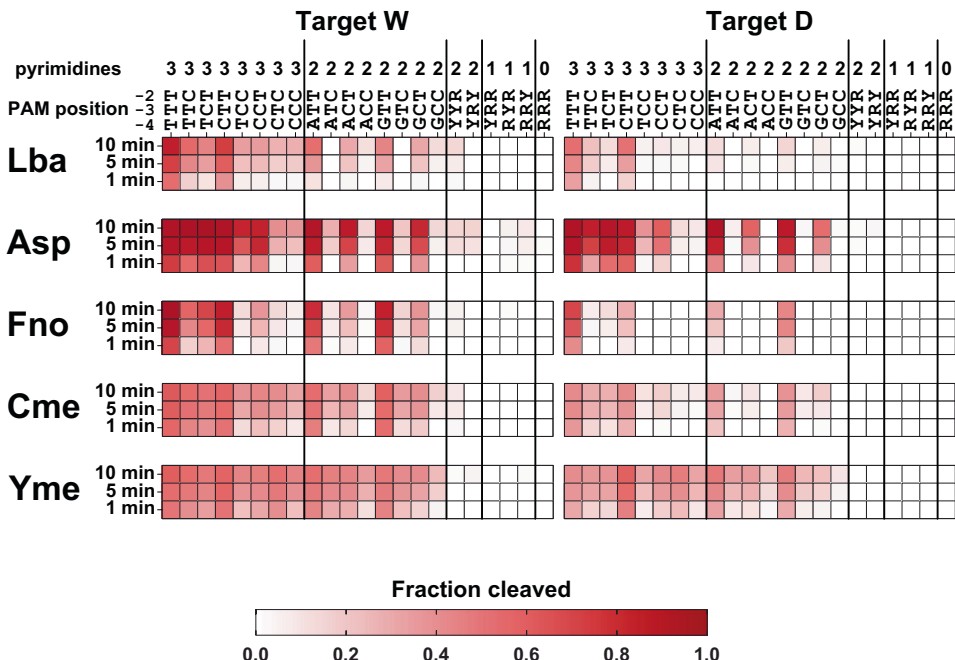

**Fig. 4 Tolerance of non-canonical PAMs by Cas12a orthologs.** Target-directed double-stranded DNA (dsDNA) cleavage activities of Cas12a RNPs were measured using in vitro assays containing fluorophore-labeled dsDNA target substrates in the presence of a 10-fold mass excess of HeLa genomic DNA. Target dsDNAs were identical except for nucleotide identity at the −2, −3, and −4 positions relative to the first nucleotide of the target sequence. Aliquots from cleavage reactions were removed and quenched after 1, 5, 10 min. Cleaved fragments were quantified and the proportion of the substrate cleaved in each condition is represented in a heatmap where the intensity of the red color corresponds to the extent of target cleavage. Reactions were performed at 37 °C for Lba, Asp, and Fno while Cme and Yme reactions were performed at 55 °C. The source data used to make the graphs can be found in Supplementary Data 2.

YmeCas12a showed a high degree of uniformity. It is notable that YmeCas12a cleavage accumulated end uniformity similar to that of the restriction enzyme controls on both strands of Target F and Target G and largely produced only a single cut between positions 17 and 18 on the NTS for all substrates except Target R. The results for Cme and YmeCas12a were identical when the cleavage experiment was performed at 55 °C instead of 37 °C (Supplementary Figure S7). We interpret these results to indicate that while the cleavage mechanism between the orthologs is expected to be highly similar, Yme represents a relatively precise variant of Cas12a.

***Trans* nuclease substrate preferences differ between Cas12a orthologs.** We examined Cme and YmeCas12a for the presence of *trans* nuclease activity and included Lba, Asp, and FnoCas12a for comparison. We performed on-target DNA cleavage with the Cas12a orthologs, corresponding crRNAs, and a 35 bp dsDNA containing a TTTC PAM sequence followed by a target sequence matching the crRNA. Subsequently, a single-stranded reporter DNA with 5′ fluorescein and 3′ Iowa Black® FQ ends was added, and fluorescence was monitored over time (Fig. 6a). When a 5 nt ssDNA randomized sequence reporter was used we observed fluorescence signal produced by Lba or AspCas12a at 37 °C that increased from the initiation of the assays and achieved maxima at ~20 min for LbaCas12a and ~10–20 min for AspCas12a. Under the same conditions, we observed minimal activity with CmeCas12a and virtually none for Fno and YmeCas12a (Fig. 6b). The signal produced by these reactions was entirely dependent upon the target DNA being present as it is required to activate the Cas12a RNP (Supplementary Figure S8).

The results were remarkably different when a 25 nt CAGT repeating sequence reporter was used (sequence in Supplementary

Data 1). The accumulation of fluorescence signal increased more rapidly and achieved maxima at ~5 min for LbaCas12a. In contrast to the limited activity seen in the 5 nt reporter assays, FnoCas12a produced a robust signal that approached maximum in ~20 min while a maximum was achieved by CmeCas12a at ~15 min or ~30 min depending on which target was used. YmeCas12 did not produce robust fluorescent signal at 37 °C with the 25 nt reporter. AspCas12a produced results that were similar with either the 5 nt or 25 nt reporter, but there was ~10–15 min difference in time that it took to achieve maximum signal depending on whether Target W or Target G was used. CmeCas12a also showed a difference in the accumulation of fluorescent signals between Target W and Target G, but the other orthologs did not. We observed a similar pattern of *trans* nuclease activity for the Cas12a orthologs when 40 nt and 6 nt CAGT repeating ssDNAs with different fluorescent labels were included together in the same reaction (Supplementary Figure S9) and when we compared polyT reporters to CAGT repeating sequence reporters (Supplementary Figure S10). In the latter case, there was a noticeable increase in activity for Lba and Asp when 5 nt polyT reporters were used instead of 5 nt CAGTC reporters, however, there was no observable difference between these reporters for the other three orthologs as they all had limited or no activity with both 5 nt reporters. Together these results are indicative of distinct differences between the *trans* nuclease activities of Cas12a orthologs with respect to target activated *trans* nuclease activity and preferences for *trans* nuclease substrates.

Surprisingly, we also observed the generation of a fluorescent signal when we substituted an RNA reporter for the DNA reporter (Supplementary Figure S11). LbaCas12a H759A which lacks crRNA processing activity[17] also produced a fluorescence signal in this assay, indicating that the RNA processing domain is

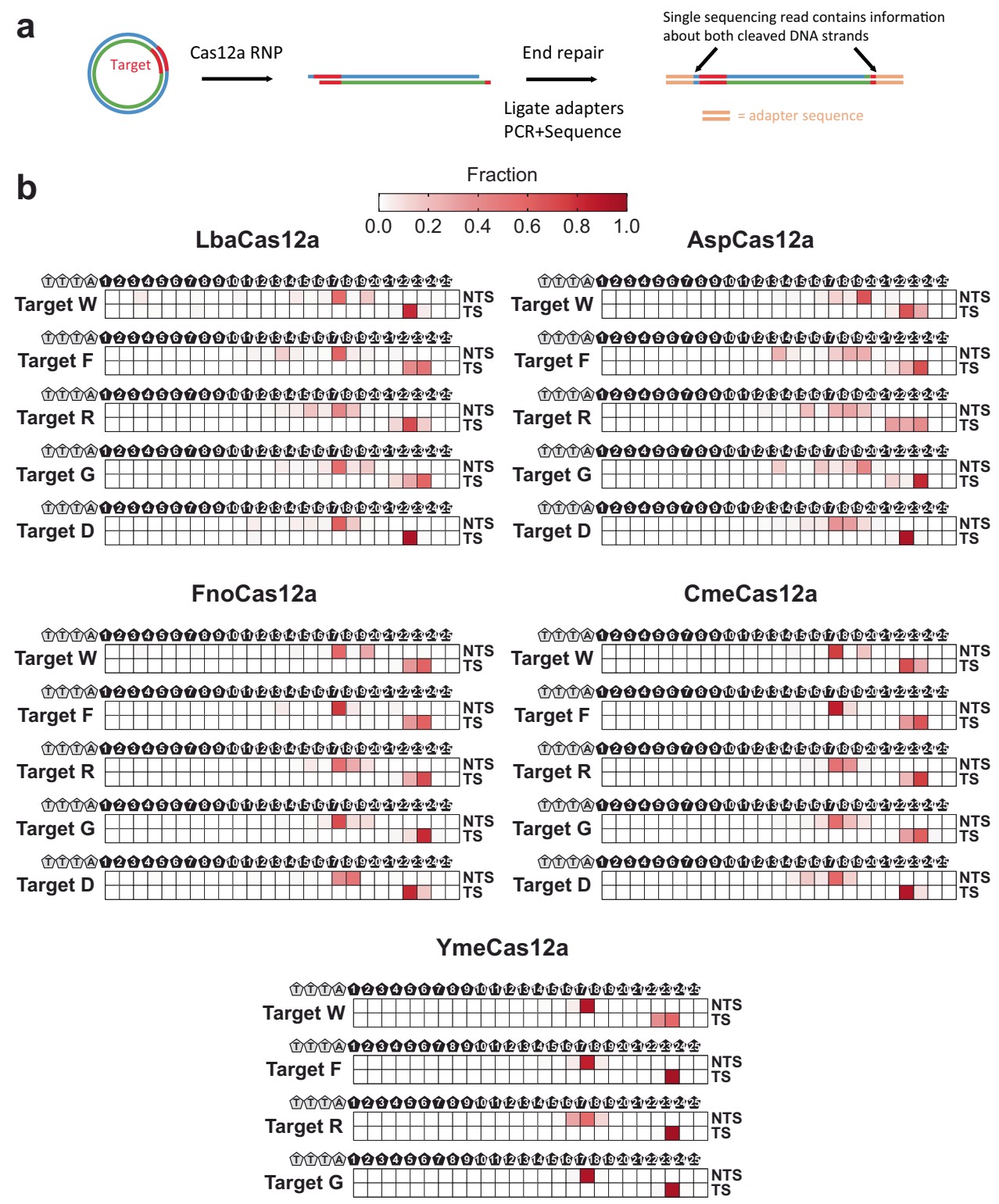

not responsible for the activity (Supplementary Figure S12). Accumulation of fluorescence signal using an RNA reporter was ~10-fold slower than when using a DNA reporter (Supplementary Figure S11B), demonstrating that while DNA is cleaved more rapidly than RNA it is possible for activated Cas12a proteins to cleave RNA in *trans*.

**YmeCas12a requires elevated temperature for optimal *trans* nuclease activity**. We were surprised that a minimal fluorescent signal was generated with Yme in reporter DNA assays after 30 min at 37 °C since it is capable of efficient target DNA cleavage at that temperature (Figs. 2 and 6b). Modulating reaction conditions, we observed an increase of fluorescence

**Fig. 5 Profile of cleavage sites generated by Cas12a orthologs. a** Schematic representation of the workflow to determine the position and abundance of cleaved DNA ends. Circular double-stranded DNAs containing a TTTA PAM sequence flanked by a target sequence were incubated with complementary crRNA-Cas12a RNPs and subjected to end repair, adapter ligation and sequencing. The substrate DNA was sequenced end to end in both reads of a paired-end sequencing run and custom scripts were used to map the cleavage site on the non-target strand (NTS) and target strand (TS). **b** The abundance of cleaved DNA ends was graphed as heatmaps representing the individual summation of all NTS cleavage position frequencies and TS cleavage position frequencies. The intensity of the red color corresponds to the fraction of ends mapped to each position on the NTS and TS. The positions are numbered 1 through 25 relative to the distance from the TTTA PAM sequence. Full data sets with matrices of the frequency of each NTS and TS cleavage site combination are available in Supplementary Data 2. Sequences of crRNAs and oligonucleotides used to make circular substrate DNAs can be found in Supplementary Data 1.

signal for YmeCas12a with a 25 nt reporter at 37 °C when NaCl concentrations in the reactions were reduced (Fig. 7). Reasoning that increased temperatures would increase this activity; we raised the reaction temperature to 55 °C. At 55 °C and 25 mM NaCl, we observed a more rapid increase in fluorescence signal for YmeCas12a. Under these conditions, the maximum signal was reached at ~10 min. Despite these optimized reaction conditions, YmeCas12a still did not degrade 5 nt reporter molecules (Fig. 7).

To assess the maximum temperature at which Yme *trans* nuclease functions, we carried out endpoint fluorescence reactions. We observed that YmeCas12a can produce *trans* nuclease activity when RNPs were incubated at temperatures up to 57.5 °C from 2-20 min before the addition of dsDNA target and 25 nt reporter substrate (Fig. 8). At 62 °C we observed that *trans* nuclease activity decreased with increasing incubation time. With a 2 min incubation of the RNP at 62 °C prior to target DNA and reporter DNA addition we observed high activity, but incubation of the RNP for 5, 10, or 20 min reduced the total fluorescence signal accumulated after addition of dsDNA target and ssDNA reporter. In contrast, for CmeCas12a, we observed that the fluorescence signal was reduced at 20 min of RNP incubation at 53.8 °C prior to the addition of dsDNA target and reporter substrate. We did not observe appreciable accumulation of fluorescence signal at 57.5 °C or higher for any time points. The temperatures at which *trans* nuclease activity is no longer observed for these orthologs correlate well with the temperature at which they lose the ability for target DNA cleavage (Fig. 2). Together, these results show that Cme and YmeCas12a are thermotolerant Cas12a orthologs with Yme being the most thermotolerant of the five orthologs studied here.

**Cas12a orthologs exhibit variable tolerance for mismatches between crRNA and target DNA sequence**. We examined differences in target recognition specificity between Cas12a orthologs by measuring *trans* nuclease activity in vitro in the presence of mismatches between crRNA and target DNA. We designed a panel of NTS DNA oligonucleotides with single and double adjacent base substitutions relative to the Target W sequence (Supplementary Data 1). The oligonucleotides were amplified with PCR to make a panel of dsDNA target substrates. The substrate panel was incubated with Cas12a RNPs in the presence of the 25 nt CAGT repeating sequence, quenched fluorescent reporter. A fully matched sequence and a completely scrambled target sequence were included in the experiment. Fluorescence was monitored over time and the time it took for each sample to reach the fluorescence value of 25% the maximum generated by the fully matched reaction is represented as a heatmap (Fig. 9).

Reactions containing substrates with single-nucleotide substitutions produced fluorescence signals at most mismatched positions throughout the target sequence over the time course of the experiment. We interpret this to indicate that single mismatches were well tolerated by the Cas12a orthologs. For Cme and YmeCas12a, reactions containing single mismatches between

crRNA and target DNA at a small number of positions failed to produce fluorescence signals over 60 min.

Substrates with double adjacent nucleotide mismatches in most cases took longer to produce fluorescence signals as compared to single mismatches, indicating that double mismatches are generally less tolerated than single mismatches. In addition, fluorescence produced in reactions containing double mismatches revealed differences between the Cas12a orthologs. LbaCas12a produced fluorescence signals in reactions with double substitutions at positions 1–8 and 18–20 but not in reactions with double mismatches at positions 9-17. In contrast, YmeCas12a produced fluorescence signals in reactions with double substitutions at positions 12–20 but not at positions 1-11. Fno and CmeCas12a show a similar profile and did not produce fluorescence signals in reactions with double mismatches at most positions. AspCas12a reactions produced fluorescence with double-mismatch substrates at all positions except for 15–18. Overall, clear differences in sensitivity for double adjacent mismatches were observed between Cas12a orthologs.

## Discussion

Cas12a proteins from Class 2 Type V-A CRISPR systems are RNA-guided DNA nucleases that have been used for genome editing in cells and DNA cleavage applications in vitro[10–12,14–16,23,24,26,28]. Most commonly, the Lba, Fno, and Asp orthologs are used for current Cas12a applications. We explored the diversity of Cas12a enzymes and identified two Cas12a orthologs of special interest in metagenomic assemblies from higher temperature biological samples; one from a **c**ompost pile **me**tagenome (Cme) and one from a **Y**ellowstone National Park hot spring **me**tagenome (Yme). As we predicted, both Cme and YmeCas12a are thermostable and thermoactive as indicated by their apoprotein state and RNP state having higher melting temperatures than Lba, Asp, and FnoCas12a (Fig. 2a) and higher target DNA cleavage activity at elevated temperatures (Fig. 2b, c), although a report has been published showing an engineered version of AspCas12a has activity up to 65 °C[53]. The activities of Yme and CmeCas12a at high temperatures compatible with isothermal amplification approaches such as LAMP, RCA, and HDA, make them of potential interest for further development toward streamlining molecular diagnostics workflows. A one-pot RT-LAMP + CRISPR reaction at 60 °C has been shown to work when a thermotolerant Cas12b protein is used[54]. Cas12b proteins are similar to Cas12a in that they are RNA-guided, DNA-targeting nucleases, however, they require a long single guide RNA (sgRNA) or a bipartite RNA duplex to function instead of a ~40 nt crRNA.

Virtually all Cas12a orthologs identified to date require a pyrimidine-rich PAM adjacent to a target sequence to carry out dsDNA targeting and cleavage[11,44–47]. We discovered subtle differences in PAM sequence requirements for in vitro assays by performing detailed comparisons of known and new Cas12a orthologs using both deep sequencing-based and single substrate approaches (Figs. 3 and 4). While either TTT or NTT appears to be the optimal sequence at positions −4 through −2 relative to the first nucleotide of the target for the five orthologs in our

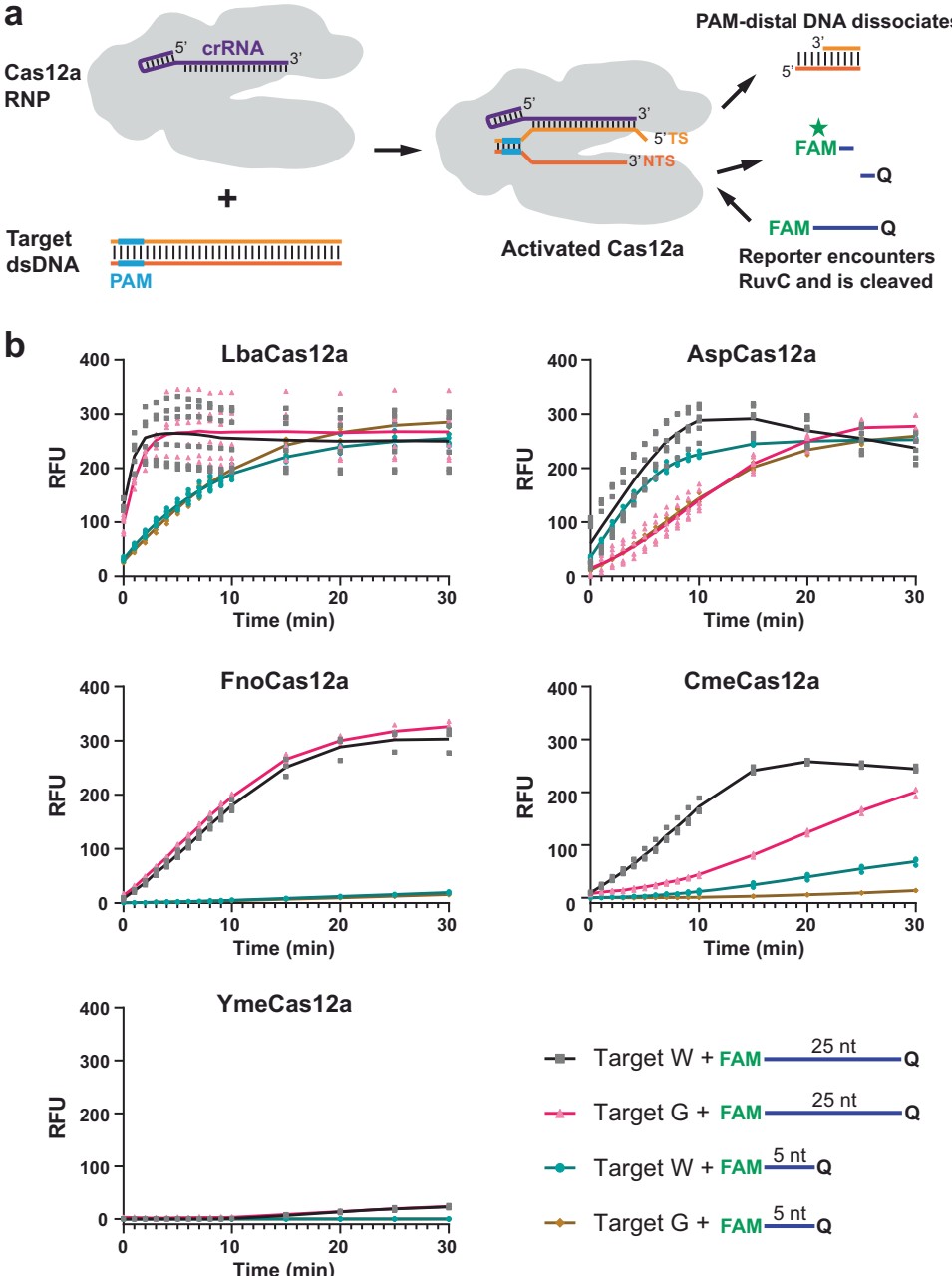

**Fig. 6 Cas12a *trans* nuclease activity is influenced by target sequence and reporter substrate length. a** Schematic representation of Cas12a *trans* nuclease activity assays. Linear double-stranded DNA containing a TTTA PAM sequence flanked by a target sequence is incubated with a complementary crRNA-Cas12a RNP at 37 °C to allow on-target cleavage. A reporter oligonucleotide with 5′ fluorophore (FAM) and 3′ quencher (Q) modifications is added and cleavage of the reporter via the *trans* nuclease activity of the activated Cas12a separates the fluorophore and quencher resulting in increased fluorescence. **b** Fluorescence was measured over time for a reporter that consisted of either a 25 nt defined sequence DNA or 5 nt randomized sequence DNA. The mean relative fluorescent unit (RFU) values for reactions with Target W DNA are plotted as black (25 nt reporter) or blue (5 nt reporter) lines. The mean RFU values for reactions with Target G DNA are plotted as magenta (25 nt reporter) or brown (5 nt reporter) lines. The RFU values represent background normalized signal derived by subtracting the fluorescence of negative control reactions that did not contain target DNA. Individual data points for all experimental replicates are plotted on the graphs and the source data can be found in Supplementary Data 2.

study, the Cas12a orthologs we tested appear to tolerate a variety of non-canonical PAM sequences, especially those with a motif of NYY (Fig. 4 and Supplementary Figure S5). These findings may be instructive for applications of Cas12a orthologs with respect to DNA sequence detection. For example, SNP or small indels that convert poorly used PAMs to more optimal PAMs could form the basis for variant detection in Cas12a-based assays.

The profile of target dsDNA ends accumulated after Cas12a RNP cleavage notably differed between the orthologs studied here (Fig. 5). Our method for assessing cleavage information using small, circular DNA substrates benefits from being able to determine the sequence on both sides of a cut DNA in both reads of a paired-end sequencing run. This provides information for both ends of a cleaved molecule rather than only a disjointed

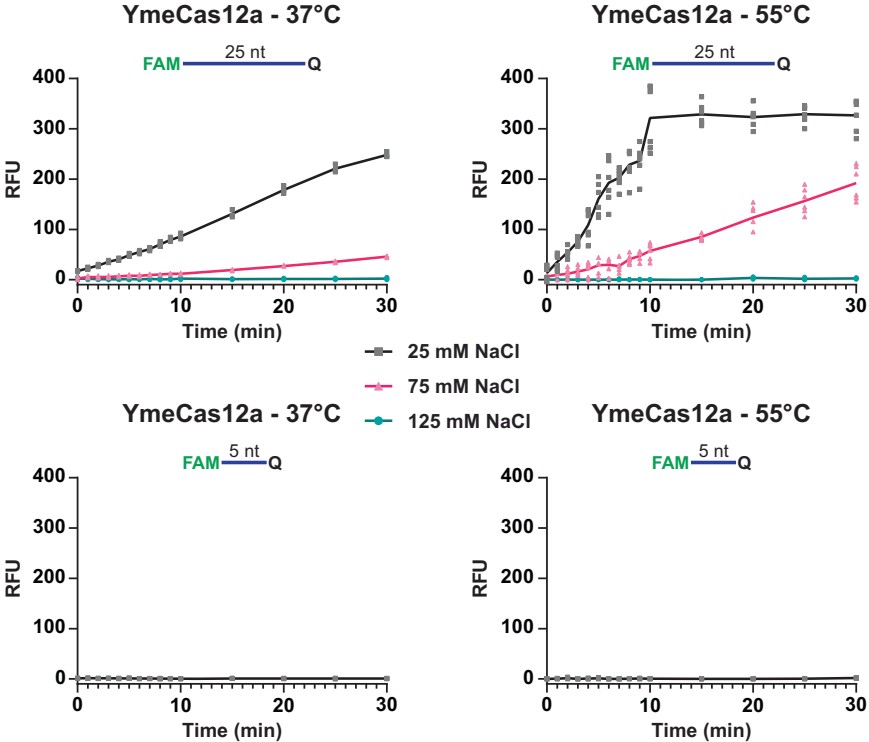

**Fig. 7 YmeCas12a requires low salt and high temperature for optimal *trans* nuclease activity.** Reactions were carried out as in Fig. 6 except that NaCl concentration was adjusted and on-target DNA cleavage plus incubation with reporter oligonucleotides was at either 37 °C or 55 °C. The mean relative fluorescent unit (RFU) values are plotted as black (25 mM NaCl), magenta (75 mM NaCl) and blue (125 mM NaCl) lines. Reactions were carried out at all three NaCl concentrations in combination with the 25 nt reporter, but only 25 mM NaCl was used in combination with the 5 nt reporter. The RFU values represent the background normalized signal derived by subtracting the fluorescence of negative control reactions that did not contain target DNA. Individual data points for all experimental replicates are plotted on the graphs and the source data can be found in Supplementary Data 2.

summation for the two sides of a cleavage target. All five orthologs showed variability in the cleavage profile depending on which target sequence was used. Our results for AspCas12a are consistent with a recent report of NTS gap formation[52], signified by multiple cleavage products with termini between positions 13 and 19. In fact, results from that study show that NTS gap formation is a precursor to target strand cleavage for AspCas12a. The other four orthologs produced more precise cleavage profiles with YmeCas12a approaching restriction enzyme-like precision on targets F and G. Our findings indicate that while gap formation is a property of AspCas12a, it does not appear to be necessary for efficient TS cleavage by all Cas12a orthologs, and that ortholog-specific differences, non-target, and target DNA sequence content, and gRNA interactions could contribute to the diversity of cleaved DNA ends. A limitation of our assay is that it is unable to distinguish between primary cleavage events and secondary cleavage event(s). This issue could potentially be addressed by performing a time-resolved assay instead of a single time point. However, it is unclear how fast a time-resolved assay would need to be to only capture primary cleavage events and it would also be limited by the unpredictable effects of *trans* nuclease activity at any time point. Despite these issues, it is clear from the heterogeneity of cleavage products in our results that using Cas12a to make predictable 5′ overhangs for cloning or other downstream applications could be problematic, but an ortholog such as YmeCas12a shows potential for generating cohesive sticky ends more consistently than others.

In addition to single-turnover target DNA cleavage in *cis*, Cas12a proteins can carry out non-specific multi-turnover degradation of ssDNA in *trans* via their RuvC domain[23–26]. This occurs because Cas12a remains in an activated state after the crRNA-target DNA duplex is formed and target-bound Cas12a RNP persists post-*cis* cleavage. Based on structural analysis it has been proposed that ssDNA should be the only substrate for *trans* nuclease activity as it can fit in the active site of the activated crRNA-FnoCas12a RNP, but dsDNA would have steric clashes[25]. Despite this, *trans* nuclease activity of LbaCas12a was reported to affect dsDNA as well[23,24,55]. It has been proposed that the partially single-stranded nature of negatively super-coiled DNA and fraying at the ends of linear dsDNA explain why dsDNA appears to be affected by *trans* nuclease activity[27]. Our work shows that there are differences between Cas12a orthologs that impact *trans* nuclease activity. We show that the tolerance of mismatches between crRNA and target DNA varies between the orthologs (Fig. 9). All orthologs produced *trans* nuclease activity with most of the single-nucleotide substitutions spanning the target sequence, although Cme and Yme showed slightly more discrimination at a few positions. This result is perhaps not entirely surprising as the lack of discrimination towards single-nucleotide substitutions has necessitated the design of guide RNAs with intentional mismatches to differentiate between sequences with single-nucleotide polymorphisms in detection assays with Cas12b proteins[56]. In contrast to single substitutions, a strong discrimination against double substitutions at many positions was observed. Surprisingly, Lba primarily tolerated double substitutions proximal to PAM region and discriminated against those distal from the PAM, whereas the opposite was true for Yme. In addition, Fno and Cme discriminated against the majority of double substitutions while Asp largely tolerated all except those between positions 15 and 18. These results highlight the underlying diversity between

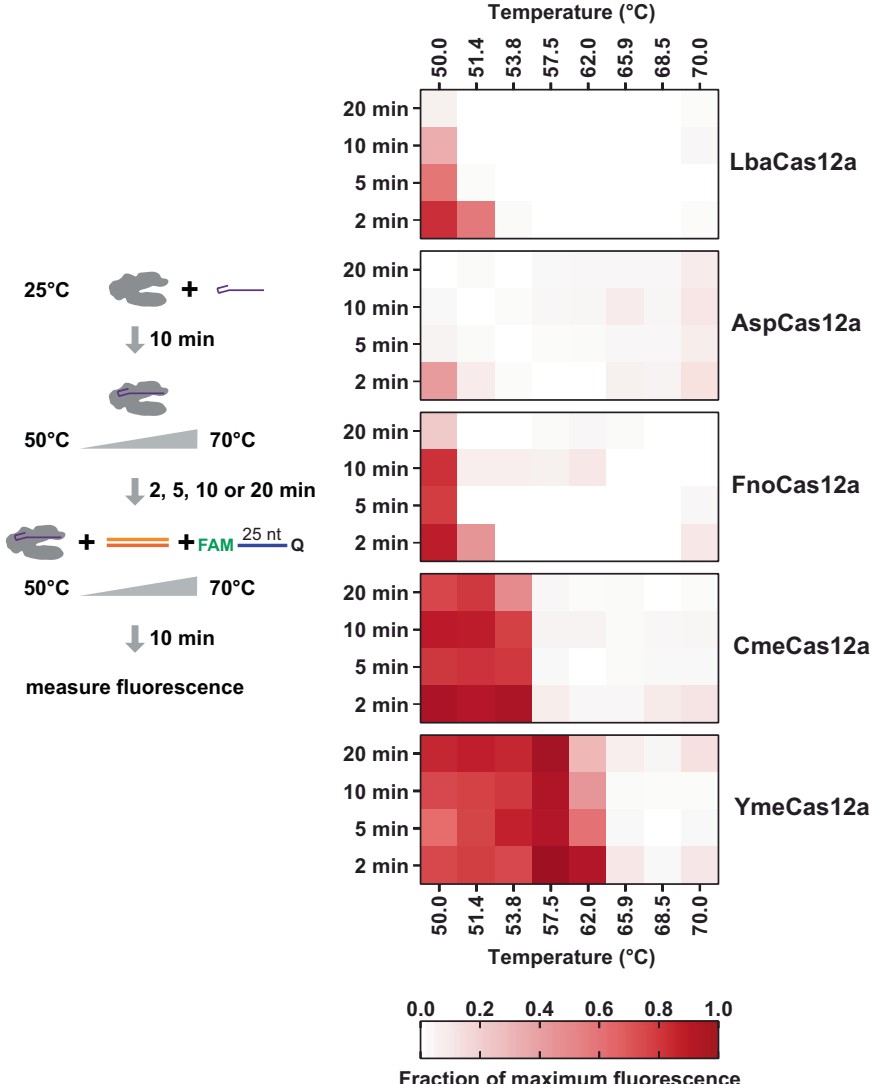

**Fig. 8 High temperature *trans* nuclease activity initiated by DNA target cleavage.** Cas12a RNPs were incubated at the temperatures indicated for either 2, 5, 10, or 20 min prior to the addition of target double-stranded DNA and a 25 nt single-stranded DNA reporter with 5′ fluorophore and a 3′ quencher modifications. Reactions were then incubated at the indicated temperature for 10 min and fluorescence was measured at the end of the reaction. The measured values were averaged and divided by the maximum value of fluorescence possible if all reporter DNA was cleaved in the assay. The intensity of the red color in the graphs corresponds to the fraction of maximum fluorescence the Cas12a enzymes were able to generate for each time and temperature combination. Source data used to generate this graph is provided in Supplementary Data 2.

Cas12a orthologs and the need to carefully design and test guide RNA-target DNA combinations for assays that utilize *trans* nuclease activity.

We also show that Lba and Asp can degrade both 5 nt and 25 nt ssDNA reporters while Fno, Cme and Yme produce very little or no activity when 5 nt reporters are used. Additionally, YmeCas12a does not cleave 5 nt reporters even under the most optimal conditions we have found (Figs. 6, 7 and S10). Surprisingly, we find that RNA is also cleaved by Cas12a *trans* nuclease activity, albeit at a rate that was typical ~10-fold less than when using ssDNA as a substrate. Generation of a fluorescent signal from the RNA reporter oligonucleotide required crRNA-Cas12a RNP and target DNA. Although RuvC domains are thought to be specific for DNA substrates, another study has shown that the RuvC domain of a Cas12g1 enzyme also has activity on RNA[57] and Cas12j utilizes its RuvC domain for RNA processing[58], suggesting the activity might be a common property of Cas12 RuvC domains. This could be a useful property for the phage

immunity function of CRISPR-Cas12 systems in their native organisms. Upon activation by a foreign target sequence, the activated Cas12 RNP might be able to degrade all single-stranded nucleic acids that are encountered. This could benefit the population of native organisms either by inhibiting phage proliferation via degrading its nucleic acids or via destroying the cellular machinery of infected cells that the phage requires for proliferation. Currently however, there is no experimental evidence that we are aware of to indicate that *trans* nuclease activity has an important biological function.

For particular in vitro applications, such as making perfect 5′ overhangs for cloning applications or for cleavage of a fluorescent ssDNA reporter oligonucleotide in *trans* to detect low abundance targets[23,24,26] it would be desirable to have the *trans* nuclease activity of the Cas12a enzyme to be as low or as high as possible, respectively. We conclude from our in vitro cleavage and *trans* nuclease activity experiments that monovalent salt is detrimental to Cas12a *trans* nuclease activity and it can be supressed by

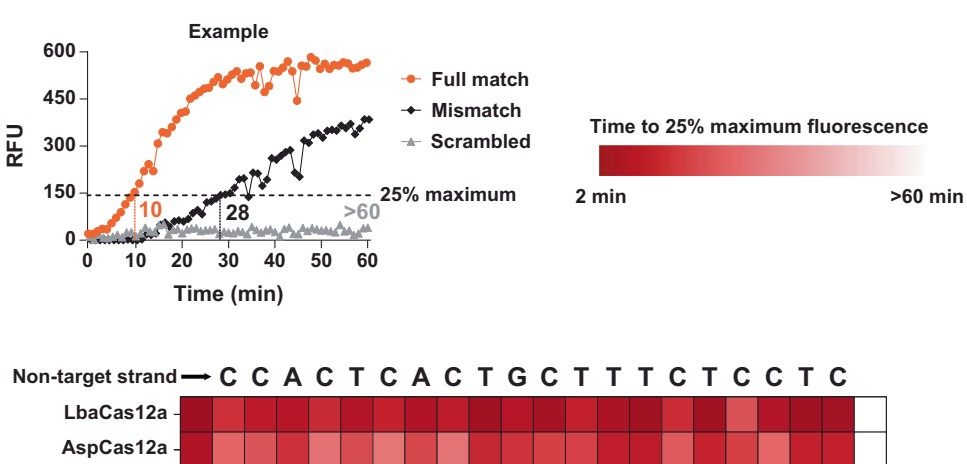

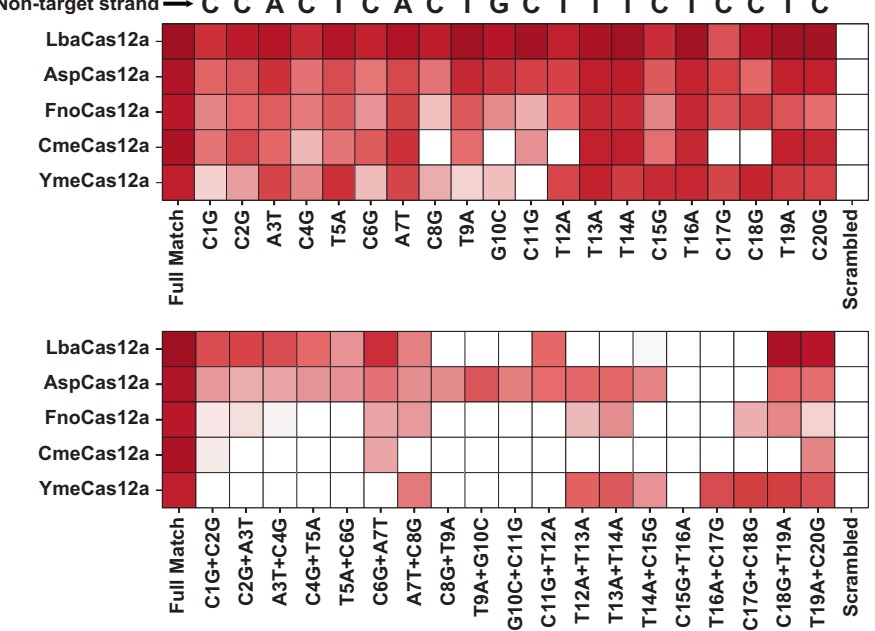

**Fig. 9 Cas12a orthologs tolerate mismatches between crRNA and target DNA for activation of *trans* nuclease activity.** *Trans* nuclease activity generated by Cas12 RNPs was measured using in vitro assays containing target DNA and a 25 nt single-stranded DNA reporter oligonucleotide with 5′ fluorophore (FAM) and 3′ quencher (Q) modifications. Target DNAs were identical in sequence except for nucleotide identity at the indicated positions relative to the first nucleotide of the non-target strand sequence. Compensatory substitutions were present in the target strand of the DNA to maintain the potential for proper base pairing between the two strands of target DNA. Fluorescence was monitored over time and recorded in 1 min increments for 60 min. The time it took for each reaction to reach 25% of the maximum relative fluorescent units (RFU) attained by the fully matched target DNA reaction was determined (see Example graph) and a heatmap was generated where the intensity of the red color corresponds to the time. Reactions were performed at 37 °C for Lba, Asp, and Fno while Cme and Yme reactions were performed at 55 °C. The source data used to make the graph can be found in Supplementary Data 2.

concentrations of NaCl at which target dsDNA cleavage is still efficient. While it is known that salt concentration can alter the stability of annealed nucleic acid strands, the $Mg^{2+}$ concentration is the driving determinant of DNA melting temperature within the range of $Mg^{2+}$ and NaCl concentration used in this study[59] and $Mg^{2+}$ concentration was kept constant throughout our experiments. Since the effect of NaCl concentration on nucleic acid duplex stability is likely negligible in our case, it could be that the effect on *trans* nuclease activity we observed was due to modulation of the electrostatic potential required for nucleic acids to interact with the activated Cas12a RNP in *trans*, i.e. in lower salt concentrations transient electrostatic interactions with DNA are more favorable than in higher salt concentrations. Although reaction conditions may need to be adjusted for given crRNA-target combinations and different Cas12a orthologs, in general, keeping reaction times short and NaCl concentration high will reduce the impact of *trans* nuclease activity while the opposite will increase it.

An important question remaining regarding *trans* nuclease activity of Cas12a is what effect it has on genome editing in cells.

Although LbaCas12a produces *trans* nuclease activity that affects ssDNA, dsDNA, and RNA, it works well in genome editing in mammalian cells and is reported as producing a very low level of off-target edits[60,61]. Additionally, LbaCas12a is the Cas12a of choice for genome editing in organisms that are grown at temperatures below 30 °C because wild-type AspCas12a has minimal activity below 30 °C[62]. Together, these successes indicate that *trans* nuclease activity does not have catastrophic or even readily noticeable effects on the genomic stability of cells[63]. Potential explanations might include removal of the RNP from the target DNA post-cleavage by cellular repair machinery before *trans* nuclease activity causes pervasive damage, protection of cellular DNA by histones and other proteins, and mutation-free repair of DNA nicks. Ultimately, it will be difficult to isolate and study the actual effect of *trans* nuclease activity on genome editing since it is reliant on the same RuvC active site as the on-target activity. Perhaps the careful timing of introducing inhibitors to turn off or reset the Cas12a RNP can be used to help distinguish the effects of on-target and *trans* nuclease activity in cells[64,65]. It will be important to support future work aimed to fully understand the

consequences of Cas12a *trans* nuclease activity in genome editing applications.

Recent studies in addition to this one have expanded the known diversity and classification of Cas12 systems and have introduced mutant Cas12a proteins with altered PAM specificities[46,57,66–68]. Future advances including detailed structures and structure-guided biochemical analysis of a variety Cas12a orthologs will provide a more complete understanding of the basis of the diversity in activity between orthologous Cas12a enzymes. A more complete understanding may enable further exciting applications of Cas12a orthologs including more specific, sensitive, and streamlined nucleic acid diagnostics or improved genome editing in a variety of organisms.

## Materials and methods

**Oligonucleotides**. All DNA and RNA oligonucleotides were synthesized by Integrated DNA Technologies (IDT, Coralville IA) or MilliporeSigma (St. Louis, MO) and the sequences are available in Supplementary Data 1. Mature crRNA sequences for Lba, Asp, FnoCas12a were obtained from a previous publication and were appended with a 20 nt targeting sequence[11]. The mature crRNA sequence for Cme and YmeCas12a were extracted from publicly available data sets described in the Cas12a purification section below. Cas12a gRNA alignments were visualized using RNAalifold[69]: http://rna.tbi.univie.ac.at/cgi-bin/RNAWebSuite/RNAalifold.cgi.

**Phylogenetic tree**. Cas12a protein sequences were obtained from NCBI (https://ncbi.nlm.nih.gov) and several publications[11,42–44]. A phylogenetic tree was generated using Geneious Prime® (https://www.geneious.com) software and an alignment type of global alignment with free end gaps. The cost matrix was Blosum62 with a genetic distance model of Jukes-Cantor and a tree build method of Neighbor-Joining.

**Cas12a purification**. EnGen® LbaCas12a from New England Biolabs (NEB #M0653, Ipswich, MA) was used for all experiments involving LbaCas12a. An engineered version of AspCas12a was obtained from Integrated DNA Technologies[70] (IDT; Alt-R® A.s. Cas12a (Cpf1) Ultra). Compost metagenome (Cme) and Yellowstone metagenome (Yme) Cas12a and the metagenomic sequences encoding them were obtained from publicly available data sets through the Joint Genome Institute's Integrated Microbial Genomes & Metagenomes resource[41] (IMG/M): https://img.jgi.doe.gov/cgi-bin/m/main.cgi. For Cme and Yme, the Gold Analysis Project IDs are Ga0079224 and Ga0078972 and the contigs from which they were recovered are 100030203 and 1022121, respectively. We obtained written permission from Jeanette Norton and Robert Kelley—the Principal Investigators of the metagenome sequencing projects to publish our findings on these enzymes. The protein sequence for FnoCas12a was obtained from the literature[11]. Fno, Cme, and YmeCas12a were purified via the following scheme. *E. coli* optimized DNA sequences encoding the proteins were cloned into plasmids with SV40 nuclear localization signals encoded on both sides of the protein sequence and a hexahistidine tag to facilitate purification. The recombinant protein was expressed in *E. coli* NiCo21 (DE3) cells (NEB #C2925) in LB media containing Kanamycin (40 μg/ml) at 37 °C (or 30 °C for Yme) until the growth reached the mid-exponential phase at which time IPTG was added to a final concentration of 0.4 mM and the temperature was shifted to 23 °C (or 16 °C for Yme) for 16 hr. Cells were harvested and disrupted by sonication prior to chromatographic purification. Recombinant protein was purified using an ÄKTA go FPLC (Cytiva/GE Healthcare, Chicago, IL) with HiTrap DEAE$^{FF}$, HisTrap$^{HP}$, and HiTrap Heparin $^{HP}$ columns (Cytiva). Fractions were pooled prior to a final chromatographic separation using a HiLoad 16/600 Superdex 200 pg column (Cytiva). Fractions were pooled, dialyzed and concentrated into storage buffer (20 mM Tris-HCl (pH 7.4), 500 mM NaCl, 1 mM DTT, 0.1 mM EDTA, 50% glycerol (v/v)) prior to storage at -20 °C until use.

**Cas12a RNP formation**. The formation of RNP was monitored using size exclusion chromatography. Cas12a proteins without crRNA (apo), crRNA without Cas12a protein, or Cas12a with an equimolar amount of crRNA (RNP) was incubated in 1× NEBuffer™ 2.1 (NEB #B7202) at room temperature for 10 min. Samples were passed through a 22 μm syringe filter (MilliporeSigma) and injected into a Superdex 200 Increase 10/300 GL column (Cytiva) using a 0.1 ml injection loop. The injected samples were passed through the column in a running buffer consisting of 20 mM Tris-HCl pH 7.5, 200 mM NaCl, 10 mM MgCl$_2$, 2 mM DTT with a constant flow rate of 0.3 ml/min on an AKTA go FPLC platform (Cytiva). Absorbance at 280 nm was monitored and 2 ml fractions were collected. For 'apo' and 'RNP' samples, fractions at peaks of absorbance were pooled and concentrated with an Amicon Ultra-4 device (MillliporeSigma) using a 100 kD molecular weight cutoff. For YmeCas12a an additional buffer exchange step to adjust the NaCl concentration to 30 mM was performed. The concentrate was incubated with 10 nM Target W DNA for 30 min at 37 °C for Lba, Asp, and FnoCas12a, or 20 min

at 50 °C and 55 °C for Cme and YmeCas12a orthologs, respectively. 25 nt single-stranded reporter DNA (sequence in Supplementary Data 1) was added and fluorescence was monitored in a SpectraMax M5 (Molecular Devices, San Jose, CA) in a time course of 30 min at 37 °C for Lba, Asp, and FnoCas12a orthologs, or 50 °C and 55 °C for Cme and YmeCas12a orthologs, respectively. A corresponding reaction lacking Target W DNA was utilized as a negative control for background signal subtraction.

**Thermal stability of Cas12a proteins**. A Prometheus nano differential scanning fluorimetry (nanoDSF) instrument was utilized to measure the thermal stability of Cas12a proteins. Protein samples were loaded into standard 10 μl capillaries (NanoTemper Technologies, München, Germany) at a concentration of 10 μM in 1× NEBuffer™ 2.1 freshly supplemented with 1 mM DTT. In addition to protein without guide RNA (apo), Cas12a RNPs were also subjected to thermal stability measurement. RNPs were formed by including 20 μM guide RNA in the experiment and incubating for 10 min at 25 °C prior to loading the capillaries. Fluorescence was monitored over a temperature range from 20 °C to 80 °C at a rate of 1 °C sec$^{-1}$ in the Prometheus instrument. The inflection point of the fluorescent curve is interpreted as the unfolding/melting point of the protein or RNP.

**Cas12a in vitro digests**. Cas12a crRNAs were heated to 65 °C, slow cooled to room temperature, and quantified by Nanodrop 2000 (ThermoFisher, Waltham MA) analysis prior to being used as a stock for in vitro digest experiments. Components of the reactions were equilibrated to the desired reaction temperature before assembly. Cas12a RNP's were formed by mixing crRNA and protein in 1.1× NEBbuffer 2.1 at a specific temperature for 10 min prior to the addition of nucleic acid substrates. The addition of nucleic acid substrate(s) reduced the buffer concentration to 1× and reactions were incubated at various temperatures and times as indicated. Reactions were quenched by the addition of a stop solution containing EDTA sufficient to ensure a 4-fold molar excess relative to Mg$^{2+}$ present in the sample and 0.2 units of Proteinase K (NEB #P8107). Quenched samples were diluted 1:10 before analysis by capillary electrophoresis[71], loading on an Agilent 2100 Bioanalyzer (Agilent Technologies, Santa Clara CA), or had tracking dye added prior to loading on a 15% TBE-Urea gel (ThermoFisher). DNA substrates used for digestion reactions were either generated by PCR using Q5® High-Fidelity 2× Hot Start Master Mix (NEB #M0494) or oligonucleotides produced by IDT.

**PAM and cleavage site pattern determination using a small, circular DNA substrate**. Circular dsDNAs 120 or 124 bp in size were produced and utilized as substrates in PAM and/or cleavage site determination assays via the following method that is also diagrammed in Figure S3. First, oligonucleotides containing the target sequence flanked by a randomized or defined PAM sequence and having 5′ PO$_4$ and 3′ OH ends were circularized with CircLigase™ ssDNA Ligase (Lucigen, Middleton, WI) using the protocol supplied by the manufacturer. Circularized DNA was concentrated and purified using a Monarch® PCR & DNA Cleanup Kit (NEB #T1030). In all, 20 pmol of the circularized DNA was incubated with 25 pmol of a complementary primer in a buffer consisting of 1× T4 DNA ligase buffer (NEB #B0202) and 40 μM dNTPs. The sample was heated to 65 °C for 30 sec followed by decreasing the temperature to 25 °C at 0.2 °C/second to allow the annealing of the primer. 6 units of T4 DNA polymerase (NEB #M0203) and 400 units of T4 DNA ligase (NEB #M0202) were added and the reaction was incubated at 12 °C for 1 h to allow second strand synthesis and ligation of the second strand to form circular, double-stranded DNA molecules. The second strand synthesis reaction was concentrated and purified with the Monarch® PCR & DNA Cleanup Kit and then eluted into 1× CutSmart® buffer (NEB #B7204) containing 1 mM ATP. 15 units each of Exonuclease V (RecBCD; NEB #M0345) and T5 exonuclease (NEB #M0363) were added to the sample and incubated at 37 °C for 45 min to remove DNA that was not fully ligated, double-stranded and circular. 0.04 units of Proteinase K (NEB #P8107) was added and the sample was incubated at 25 °C for 15 min prior to being concentrated and purified using a Monarch® PCR & DNA Cleanup Kit. After elution, the yield of circular DNA product was assessed using an Agilent 2100 Bioanalyzer, and the product was then used as a substrate in assays as follows. Cas12a RNPs were formed by incubating 2 pmol of guide RNA with 1 pmol of Cas12a protein in 1.1× NEBuffer™ 2.1 (NEB #B7202) at room temperature for 10 min. RNPs were added to 0.2 pmol of circular DNA substrate and incubated for 5 min at 37 °C or 55 °C depending on the experiment and then reactions were quenched by the addition of 0.04 units Proteinase K and EDTA at a final concentration of 32 mM. Control reactions to assess the composition of the randomized PAM region in the substrates were performed by digesting with BstXI (NEB #R0113) for 30 min at 37 °C. Substrates were also digested with HhaI (NEB #R0139), HinP1I (NEB #R0124), or FspI (NEB #R0135) to assess the ability of the method to identify 3′ overhang, 5′ overhang, and blunt cleavage sites, respectively. Reactions were concentrated and purified with a Monarch® PCR & DNA Cleanup Kit and the entire elution volume was used as a substrate for Illumina sequencing library construction using a NEBNext® Ultra™ II DNA Library Prep Kit for Illumina® (NEB #E7645) following the protocol provided with the kit. Either between seven and eight cycles (for restriction enzyme digested samples) or 15–18 cycles

(for Cas12a digested samples) of PCR were used to add the Illumina priming sequences plus index barcodes and then the concentration of each library was assessed on an Agilent 2100 Bioanalyzer. Libraries were pooled and sequenced on either an Illumina NovaSeq or NextSeq instrument with 2 × 150 paired-end sequencing runs. PAM and cleavage site determination and visualization was carried out using custom scripts, available in GitHub[72], which used the following position weight equation to determine the enrichment or depletion of each base at each position in the PAM preference assays:

$$score = \log_2 (Cas12a\ sample\ frequency / BstXI\ sample\ frequency) \qquad (1)$$

**Fluorescent reporter assay for _trans_ nuclease activity**. All Cas12a RNPs were formed as described above (Cas12a in vitro digests) and reactions were carried out in 1× NEBuffer™ 2.1 or recreation of NEBuffer™ 2.1 with altered levels of NaCl as indicated. Enzyme and crRNA were incubated at room temperature for 10 min at a 1:1 molar ratio and a final concentration of 50 nM or 250 nM to form RNPs. For Fig. 9 the molar ratio was 1:2 and the final concentration of enzyme was 10 nM. If the RNP was to be incubated at a higher temperature, it would then be transferred to the designated temperature for 5 min. dsDNA containing the target sequence was prewarmed to the designated temperature and added to the assembled RNPs at a final concentration of 2.5 nM, 5 nM, or 25 nM in a final reaction volume of 16 μl, 20 μl, or 95 μl. The reaction was incubated at the designated temperature for at least 10 min to ensure complete cleavage of the target DNA, which was confirmed by removal of 10 μl of the reaction (in the case of the 95 μl reaction) followed by purification using the Monarch® PCR & DNA Cleanup Kit and loading of the sample onto an Agilent 2100 Bioanalyzer. For assessment of _trans_ nuclease activity, either 2 pmol or 50 pmol of ssDNA or RNA reporter (sequences in Supplementary Data 1) was added to the reaction in a black, clear-bottom 384-well or 96-well plate (Costar #3762, #3615) at a final concentration of 100 nM or 500 nM. Fluorescence intensity ($\lambda_{ex}$: 485 nm; $\lambda_{em}$: 520 nm) was recorded with a 10–15 sec lag time between sample mixing and the first reading via an M2e or M5 SpectraMax plate reader (Molecular Devices) at regular time intervals. For Fig. 9 the target DNA and reporter were added at the same time after RNP formation and incubation of the RNP for 5 min in the presence of 16 ng HeLa genomic DNA. For Supplementary Figure S11, data points from the first 5 min were used to calculate the rate of _trans_ nuclease activity using KaleidaGraph software (Synergy Software) and a linear regression formula:

$$y = m_1 x + m_0 \qquad (2)$$

where $m_0$ is the y-intercept and $m_1$ is the slope (rate; min$^{-1}$) of the increase in fluorescent signal.

To establish the maximum possible signal generated by each reporter oligonucleotide (positive control), fluorescence measurements were also taken with reactions in which the reporter oligonucleotide was digested with 2000 units of micrococcal nuclease (NEB# M0247) instead of Cas12a (Supplementary Figure S8). Samples that lacked the target DNA complimentary to the guide RNA were included in each assay with Cas12a RNP as a negative control. Average baseline relative fluorescent units (RFU) from negative controls were subtracted from the RFU values generated by all experimental samples.

**Fluorescent endpoint reporter assay for RNP stability**. Cas12a RNPs were formed as described above for Cas12a in vitro digests and reactions were carried out in 1× NEBuffer™ 2.1 without NaCl (10 mM Tris-HCl pH 8.0, 10 mM MgCl₂, 100 ug/ml BSA). The final concentration of NaCl in the reaction was 25 mM due to the storage buffer of Cas12a. Buffer, enzyme, and crRNA were incubated at room temperature for 10 min at a 1:1 molar ratio and a final concentration of 250 nM. Cas12a RNP's were aliquoted to 8-tube strips to span a temperature gradient from 50 °C to 70 °C in a thermocycler. In a separate 8-tube strip, a 35 bp dsDNA target and 25-nt DNA reporter were combined. RNP and target/reporter were separately incubated at the temperature gradient for 2, 5, 10, and 20 min, at which point they were combined. The final concentration of target and reporter were 25 nM and 500 nM, respectively. After an additional 10 min of incubation, a stop solution was added (final concentration of 40 mM EDTA, 0.2× NEBuffer™ 2.1, and 0.4 U Proteinase K). The stopped reactions were transferred to a black, clear-bottom 384-well plate (Costar #3762) and fluorescence intensity ($\lambda ex$: 485 nm; $\lambda em$: 520 nm) was read via an M2e plate reader (Molecular Devices, San Jose, CA).

**Statistics and reproducibility**. All experiments were reproducible as indicated by the source data provided in Supplementary Data 2. Graphs in Figs. 6, 7, S1, S8–12 show individual data points representing all experimental replicates. Most reactions in these experiments were repeated a total of three times and up to six times although a few reactions only had two replicates. Experiments shown in Figure S1 were repeated twice. Two-way ANOVA with Tukey's multiple comparisons test was used to determine if the rates for DNA reporters are significantly different than RNA reporters in Figure S11.

**Reporting summary**. Further information on research design is available in the Nature Research Reporting Summary linked to this article.

## Data availability
Data from experiments involving high-throughput sequencing have been deposited in the NCBI Sequence Read Archive (https://www.ncbi.nlm.nih.gov/sra) under accession number PRJNA718241. Source data for other experiments can be found in Supplementary Data 2 and all other data are available from the corresponding author upon reasonable request.

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

## Acknowledgements

We thank Jeanette Norton and Robert Kelley—the Principal Investigators of the meta-genome sequencing projects that contained contigs from which Cme and YmeCas12a were identified—for their permission to publish our findings on these enzymes. We also thank Juan Pan and Paul Yourik with advice on data analysis and Laurie Mazzola, Kristen Augulewicz, Harold Bell, and Danielle Fuchs for assistance with Illumina sequencing and capillary electrophoresis. Funding was provided by New England Biolabs, Inc.

## Author contributions

R.T.F. and G.B.R. conceived the project and wrote the manuscript. P.R.W. and G.B.R. identified Cas12a candidates in metagenomic databases. M.M. and A.N. identified Cas12a candidates and purified Cas12a proteins for the study. R.T.F., J.L.C., and M.M. performed the experiments. Z.S. provided bioinformatic support and wrote custom scripts for data analysis.

## Competing interests

All authors are current or former employees in the Research Department of New England Biolabs Inc. New England Biolabs is a manufacturer and commercial supplier of molecular biology reagents, including enzymes and buffers used in this study. Cme and YmeCas12a are not commercial products. The authors' affiliation does not affect their impartiality, adherence to journal standards and policies, or availability of data.
