## [Peer Review File · Communications Biology]

Reviewers' comments:

Reviewer #1 (Remarks to the Author):

The major claim of the paper is that the newly tested Cas12a variants (Cme and Yme) are active at higher temperatures than the more widely used Cas12a enzymes.

I feel the authors presented sufficient evidence to support this claim using well controlled experiments.

This study is novel in the aspect that it provides new characterization of these two Cas12a variants. This study will likely be of interest to the community and will help set a foundation for using these variants in vivo or for additional in vitro applications. However, I do not think this study will have a significant impact in a wider field as the study is narrowly focused on genome modifying variants that are active at a slightly higher temperature (~10 degrees C when forming RNPs at 25C) than other widely used enzymes. The authors could do a better job highlighting how this modest difference can significantly impact the field.

Overall, the paper is well-written and provides convincing work. With only a minor reservation of the overall impact the paper will make, I have no issues or comments regarding the experimental design or results.

Reviewer #2 (Remarks to the Author):

Ryan T. Fuchs and colleagues identify and characterize two distinct thermophilic Cas12a orthologs from metagenomic sources. The authors show that the novel type V enzymes are thermotolerant and benchmark their in vitro activity, relative to three mesophilic and well characterized Cas12a enzymes (Lb, Fn and As). Most biotechnologically relevant characteristics are thoroughly investigated (PAM recognition, cutting activity at different temperatures, cleavage pattern). Overall, the authors confirm previous observations on Cas12a proteins and add two similar orthologs to the set of available enzymes. The findings are of limited interest for the CRISPR field and diagnostics/gene editing community.

The study is well executed and I appreciate the characterization of the thermotolerant enzymes along the three mesophilic variants. PAM requirements and DNA cleavage patterns are elegantly analyzed in vitro. I believe that the thorough characterization warrants publication in Communications Biology if the authors are able to address the following comments and suggestions.

1) Line 48ff, the authors introduce Cas9 in detail. The details are of limited relevance for this study, as the authors exclusively characterize Cas12a variants and do not discuss their findings in context of Cas9. I believe that the manuscript would benefit from a shorter introduction of Cas9 in favor of a more detailed introduction focussing on thermotolerant Cas enzymes, their application and potential. There are ample references for the application of thermotolerant Cas12 proteins for single pot nucleic acid detection methods (e.g. <https://www.nejm.org/doi/10.1056/NEJMc2026172>). The authors could also consider introducing the differences in Cas12 enzymes and how those differences impact applications. For instance, thermotolerant Cas12b does require a long sgRNA, or tracrRNA:crRNA duplex to function, while thermotolerant Cas12a would only require a short crRNA guide (reducing the cost of production, etc.).

2) Line 49, the authors mention maturation of crRNA. The paragraph, if e.g. rewritten for Cas12b, should introduce the concept of CRISPR array transcription and precursor (pre)-crRNA processing to understand why maturation is required (and what the substrates are).

3) Line 56, consider using pre-crRNA instead of primary transcript.

4) Line 65, Cme and Yme are mentioned. Please explain the abbreviations here. Are there any hints on the species that encode for the enzymes?

- 5) Line 67f, the list would benefit from primary citations for the respective orthologs.
- 6) Line 233f, citations for the metagenomic data is missing. There have been recent controversies on the public release of metagenomic data by the JGI. Please confirm that the authors adhere to the policies described here: <https://jgi.doe.gov/user-programs/pmo-overview/policies/#data-util>
- 7) Line 243, the authors state that they identified the guide RNA sequences from metagenomes. I believe that the data are DNA based (not metatranscriptomics). Please reword accordingly. A minimal supplementary figure showing the genomic locus and CRISPR array might be helpful for the inexperienced reader to understand the composition of the locus/ nature of the data.
- 8) Line 245, the authors mention that the crRNA has a characteristic stem loop structure. Please provide a reference for this statement.
- 9) Line 249, replace "isolated" with "identified in".
- 10) Line 252, the authors analyze the RNP. This paragraph would benefit from a short description of the used crRNA. How was the RNA designed. Is Cas12a-mediated processing required to form the active RNP? A control assessing the successful/efficient formation of the RNPs is lacking. A simple SEC chromatogram is sufficient to demonstrate the formation.
- 11) Line 261-268, consider moving this paragraph to the method section.
- 12) Line 304, the authors test the nuclease activity on cognate targets in presence of a 10-fold excess of HeLa genomic DNA. This seems to be a high fraction of target DNA (considering the actual molarity of the target site), not present in any actual 'real-life' application. Please explain why this concentration of HeLa genomic DNA was used.
- 13) Line 332, the sentence would benefit from a clearer explanation. It is hard for the inexperienced reader to understand that the method allows for a precise mapping of the termini of ssDNA fragments . The method is a neat addition to the available methods for cleavage pattern mapping, however, it does not allow for the mapping of a dsDNA cleavage pattern of an individual dsDNA substrate molecule (rather the mapping of ssDNA, which are produced during the illumina sequencing)
- 14) Line 356, please consider to tone down "significant differences in cleavage mechanism" - the mechanism is anticipated to be identical, given the high similarity of the enzymes and the conserved RuvC active site. The products are slightly different, while the cleavage mechanism is very likely the same.
- 15) Line 380, the authors use a specific reporter DNA. Can the authors comment on the design. E.g. Why haven't the authors decided to use a long poly-T reporter, which has been shown to function well for certain type V enzyme based trans cleavage assays? (Fig. 4, <https://science.sciencemag.org/content/362/6416/839.abstract>).
- 16) Line 460, this effects might also be explained by a certain degree of DNA sequence specificity for the NTS.
- 17) Line 480, quite interestingly, this behavior correlates to the trimming observed in figure 5b. I believe this is worth mentioning.
- 18) Line 488, RNA cutting is the ancient function of the RNaseH fold, from which the RuvC evolved. In addition, Cas12j (CasPhi) has been shown to process pre-crRNA using the RuvC active site, which could be mentioned here (<https://science.sciencemag.org/content/369/6501/333/tab-figures-data>).
- 19) Line 492, the trans cleavage activity is discussed in context of a hypothetical biological function. This has been previously hypothesized on several occasions, however, evidence is lacking

completely. It seems highly unlikely that activated Cas12a ever encounters unprotected ssDNA in a cell (other than the ssDNA produced during target identification).

20) Line 512, there is no evidence that the trans activity has any impact for genome editing (likely no ssDNA accessible for the protein). Please consider discussing:

<https://link.springer.com/article/10.1007/s13238-021-00824-z>

Reviewer #3 (Remarks to the Author):

Comments:

1. Whether the Cme and Yme - Cas12a exhibit any off-target activities? Comparative experiments using at least LbaCas12 are required, as it's important for the potential application in gene editing or detection.
2. How about the specificity of Cme and Yme? Are they sensitive to SNP?
3. In trans cleavage assay, why chosen a 25nt ssDNA/RNA reporter? Different length probes should be tested to determine the most efficient one.
4. The Yme showed relatively precise cleavage, is there base bias specific cleavage? Such as tolerance to high G/C content?
5. Figure 6B, the initial fluorescence values (0 min) are inconsistent, both the 5nt and 25nt reporters, especially in the LbaCas12a. Moreover, the 25nt ssDNA reporter showed a higher initial value, is that due to the high background signal?

Reviewer #1 (Remarks to the Author):

The major claim of the paper is that the newly tested Cas12a variants (Cme and Yme) are active at higher temperatures than the more widely used Cas12a enzymes.

I feel the authors presented sufficient evidence to support this claim using well controlled experiments.

This study is novel in the aspect that it provides new characterization of these two Cas12a variants. This study will likely be of interest to the community and will help set a foundation for using these variants in vivo or for additional in vitro applications. However, I do not think this study will have a significant impact in a wider field as the study is narrowly focused on genome modifying variants that are active at a slightly higher temperature (~10 degrees C when forming RNPs at 25C) than other widely used enzymes. The authors could do a better job highlighting how this modest difference can significantly impact the field.

Thank you for taking the time to review our paper and for your feedback. We believe that the identification and characterization of Cas12a orthologs with activity at higher temperatures will be very important when developing assays combined with target amplification technologies, such as LAMP which are typically performed at temperatures >55C. We have added new text to this effect in the introduction [lines 54-67] and discussion [lines 496-503] to better highlight our view of this significance, especially with potential applications to molecular diagnostics.

Overall, the paper is well-written and provides convincing work. With only a minor reservation of the overall impact the paper will make, I have no issues or comments regarding the experimental design or results.

Reviewer #2 (Remarks to the Author):

Ryan T. Fuchs and colleagues identify and characterize two distinct thermophilic Cas12a orthologs from metagenomic sources. The authors show that the novel type V enzymes are thermotolerant and benchmark their in vitro activity, relative to three mesophilic and well characterized Cas12a enzymes (Lb, Fn and As). Most biotechnologically relevant characteristics are thoroughly investigated (PAM recognition, cutting activity at different temperatures, cleavage pattern). Overall, the authors confirm previous observations on Cas12a proteins and add two similar orthologs to the set of available enzymes. The findings are of limited interest for the CRISPR field and diagnostics/gene editing community.

The study is well executed and I appreciate the characterization of the thermotolerant enzymes along the three mesophilic variants. PAM requirements and DNA cleavage patterns are elegantly analyzed in vitro. I believe that the thorough characterization warrants publication in Communications Biology if the authors are able to address the following comments and suggestions.

Thank you for your thoughtful and thorough review. We have addressed your specific comments as detailed below which we believe has improved our manuscript.

1) Line 48ff, the authors introduce Cas9 in detail. The details are of limited relevance for this study, as the authors exclusively characterize Cas12a variants and do not discuss their findings in context of Cas9.

I believe that the manuscript would benefit from a shorter introduction of Cas9 in favor of a more detailed introduction focussing on thermotolerant Cas enzymes, their application and potential. There are ample references for the application of thermotolerant Cas12 proteins for single pot nucleic acid detection methods. The authors could also consider introducing the differences in Cas12 enzymes and how those differences impact applications. For instance, thermotolerant Cas12b does require a long sgRNA, or tracrRNA:crRNA duplex to function, while thermotolerant Cas12a would only require a short crRNA guide (reducing the cost of production, etc.).

Thank you for your helpful feedback. We have adjusted the introduction to have more focus on the applications of Cas12a and reduced our description of Cas9.

2) Line 49, the authors mention maturation of crRNA. The paragraph, if e.g. rewritten for Cas12b, should introduce the concept of CRISPR array transcription and precursor (pre)-crRNA processing to understand why maturation is required (and what the substrates are).

We thank the reviewer for the helpful comment. We have revised this section to remove mention of crRNA maturation.

3) Line 56, consider using pre-crRNA instead of primary transcript.

We've removed mention of primary transcripts along with discussion of crRNA maturation.

4) Line 65, Cme and Yme are mentioned. Please explain the abbreviations here. Are there any hints on the species that encode for the enzymes?

We have now amended the introduction to explain the Cme and Yme abbreviations [lines 68-69]. The species that encode for these enzymes have not been characterized/fully sequenced and come from a 7.8 Kb contig named 100030203 (Cme) and a 7.9 Kb contig named 1022121 (Yme) from the metagenomic databases indicated in the citation [lines 106-107].

5) Line 67f, the list would benefit from primary citations for the respective orthologs.

We have updated the current version of the manuscript to include primary citations for the orthologs [line 72]

6) Line 233f, citations for the metagenomic data is missing. There have been recent controversies on the public release of metagenomic data by the JGI. Please confirm that the authors adhere to the policies described here:

We have now added the citation to the results section in addition to the methods section. We have contacted the Principal Investigators from the two sets of metagenomic data by email and received their permission to publish using the Cme and Yme sequences in this study. We have added citations for the metagenomic data to the results section [line 265] and acknowledgements to the metagenome study PIs in the acknowledgements section [line 624-626].

7) Line 243, the authors state that they identified the guide RNA sequences from metagenomes. I

believe that the data are DNA based (not metatranscriptomics). Please reword accordingly. A minimal supplementary figure showing the genomic locus and CRISPR array might be helpful for the inexperienced reader to understand the composition of the locus/ nature of the data.

Thank you for the suggestion, we have reworded this description and the revised text can be found in lines 275-276. We have added a graphical depiction of the loci to Figure 1.

8) Line 245, the authors mention that the crRNA has a characteristic stem loop structure. Please provide a reference for this statement.

We have added a reference for this statement which can now be found at lines 277-278.

9) Line 249, replace “isolated” with “identified in”.

We have revised the text based on this suggestion and it can now be found at line 281.

10) Line 252, the authors analyze the RNP. This paragraph would benefit from a short description of the used crRNA. How was the RNA designed. Is Cas12a-mediated processing required to form the active RNP? A control assessing the successful/efficient formation of the RNPs is lacking. A simple SEC chromatogram is sufficient to demonstrate the formation.

We have expanded this paragraph’s description of the crRNA. The description can be found at lines 282-284. We have added new data in a new supplementary figure (S1) that shows SEC traces of apo-Cas12a proteins, crRNAs and Cas12a + crRNA RNPs. The new data are highlighted in the results section on lines 287-288.

11) Line 261-268, consider moving this paragraph to the method section.

We removed these lines in the process of revising the manuscript. As suggested, the content can be found in the “Cas12a *in vitro* digests” methods section starting on line 156.

12) Line 304, the authors test the nuclease activity on cognate targets in presence of a 10-fold excess of HeLa genomic DNA. This seems to be a high fraction of target DNA (considering the actual molarity of the target site), not present in any actual ‘real-life’ application. Please explain why this concentration of HeLa genomic DNA was used.

The purpose of the HeLa DNA was to demonstrate that “background” non-target DNA slows the ability of Cas12a to cleave matched-target DNA and visibly increases the contrast in the heat map graphs between optimal and sub-optimal PAMs (compare Figure 4 to Figure S5). Our intent was not to mimic cellular/nucleus conditions. Rather, this condition was intended to mimic an *in vitro* digest of a sample in a molecular diagnostic test. In this case, there would be some background DNA from the sample and amplified DNA containing the target. We have added text on line 336 clarifying this.

13) Line 332, the sentence would benefit from a clearer explanation. It is hard for the inexperienced reader to understand that the method allows for a precise mapping of the termini of ssDNA fragments .

The method is a neat addition to the available methods for cleavage pattern mapping, however, it does not allow for the mapping of a dsDNA cleavage pattern of an individual dsDNA substrate molecule (rather the mapping of ssDNA, which are produced during the illumina sequencing)

We have rewritten the text to clarify the explanation on lines 364-367. In addition, we have modified Figure 5 to make the diagram more clear.

The DNA substrate used in these assays is circular and short enough that both sides of the cleavage site are present on the same linear, double-stranded molecule after cleavage. The end repair step during library construction encodes information about the overhang that was generated by the cleavage event on both strands and ends of the library molecule. Thus, even though the library molecule is made into single stranded molecules during the sequencing process, those single stranded molecules encode information about the double stranded cleavage event.

14) Line 356, please consider to tone down “significant differences in cleavage mechanism” - the mechanism is anticipated to be identical, given the high similarity of the enzymes and the conserved RuvC active site. The products are slightly different, while the cleavage mechanism is very likely the same.

Thank you for this feedback, we have reworded this statement. We agree that the overall mechanism is likely highly similar, and the reworded sentence can be found on lines 391-392.

15) Line 380, the authors use a specific reporter DNA. Can the authors comment on the design. E.g. Why haven't the authors decided to use a long poly-T reporter, which has been shown to function well for certain type V enzyme based trans cleavage assays? (Fig. 4, <https://science.sciencemag.org/content/362/6416/839.abstract>).

We have added additional new data that are shown in new supplementary figure (S10). In this figure we compare poly-T to CAGT repeat sequence reporters of 5 and 25 nt in length. While we do observe a large difference for Lba and Asp with 5 nt length reporters of different sequence, the poly-T sequence does not rescue the inability of Fno, Cme and Yme to use 5 nt length reporters effectively. These data are commented on in the results section – lines 418-422 and in the discussion – lines 560-563.

16) Line 460, this effects might also be explained by a certain degree of DNA sequence specificity for the NTS.

Thank you for this comment. We agree that it is unclear how sequence content in the target region influences Cas12a cut site(s). We have expanded this sentence to incorporate this potential explanation. The revised statement can be found at lines 527-528.

17) Line 480, quite interestingly, this behavior correlates to the trimming observed in figure 5b. I believe this is worth mentioning.

We have removed this sentence about AspCas12a. Our new results in figure S10 are not consistent with the statement. Taking all of our data into consideration, we are not able to make or explain a correlation between trimming after on-target cleavage and reporter substrate preference. For instance, Lba and

Fno have very similar cleavage / trimming patterns as we show in Figure 5 but vary in their reporter substrate preferences. We interpret this to mean that either there is no strong correlation between the two activities, or that we would more data to come to establish a correlation.

18) Line 488, RNA cutting is the ancient function of the RNaseH fold, from which the RuvC evolved. In addition, Cas12j (CasPhi) has been shown to process pre-crRNA using the RuvC active site, which could be mentioned here.

Thank you for pointing this out, we've added the reference. The revised text can be found at lines 567-568 in the revised manuscript.

19) Line 492, the trans cleavage activity is discussed in context of a hypothetical biological function. This has been previously hypothesized on several occasions, however, evidence is lacking completely. It seems highly unlikely that activated Cas12a ever encounters unprotected ssDNA in a cell (other than the ssDNA produced during target identification).

We have revised the text to clarify our viewpoint on any potential biological function for trans nuclease activity. The revised text can be found in lines 574-575 in the manuscript.

20) Line 512, there is no evidence that the trans activity has any impact for genome editing (likely no ssDNA accessible for the protein). Please consider discussing:

We have revised the text and added a reference for lack of impact on genome editing. The revised text at lines 574, 575, and 598 have clarified our perspective on the impacts of trans nuclease activity on genome editing.

Reviewer #3 (Remarks to the Author):

Thank you for your helpful review. We have addressed your specific comments as detailed below which we believe has improved our manuscript.

Comments:

1. Whether the Cme and Yme - Cas12a exhibit any off-target activities? Comparative experiments using at least LbaCas12 are required, as it's important for the potential application in gene editing or detection.

2. How about the specificity of Cme and Yme? Are they sensitive to SNP?

We thank the reviewer for raising these related questions. In our revised manuscript we have added substantial new data to address the off-target and mismatch tolerance of Yme and CmeCas12a RNPs in comparison to Lba, Asp, and FnoCas12a RNPs. The data are summarized in figure 9 and detailed in the Results section on lines 457-485 and discussion on lines 546-559. In brief, we tested the activity of the 5 Cas12aRNPs on a panel of target DNAs with single or double adjacent mismatches. We report mismatch sensitivity by monitoring *trans* nuclease activity which depends on *cis*-target cleavage and is more

important to *in vitro* detection of DNA sequences which we feel is the exciting and likely application of Cme and Yme Cas12a.

To assess off-target activities in the context of genome editing would require an organism or cell type that is viable at a broad temperature range e.g. 37-65 °C encompassing the optimal temperature ranges of the Cas12a enzymes that we studied. Our preliminary attempts at genome editing with Yme in human cell lines failed, likely because Yme has relatively poor activity at 37 °C, precluding comparison with mesophilic orthologs Lba, Asp, and FnoCas12a.

3. In trans cleavage assay, why chosen a 25nt ssDNA/RNA reporter? Different length probes should be tested to determine the most efficient one.

We have added additional new data that are shown in a new supplementary figure (S10). In this figure we compare poly-T to CAGT repeat sequence reporters of 5 and 25 nt in length. While we do observe a large difference for Lba and Asp with 5 nt length reporters of different sequence, the poly-T sequence does not rescue the inability of Fno, Cme and Yme to use 5 nt length reporters effectively. The data are commented on in the results section – lines 418-422 and in the discussion – lines 560-562.

4. The Yme showed relatively precise cleavage, is there base bias specific cleavage? Such as tolerance to high G/C content?

Yme shows a higher degree of precision overall versus the other Cas12a orthologs for the targets that we have tested. However, we are unable to ascertain the extent of base bias specific cleavage, if any. Three of the targets (W,G,R) used in our study had GC content between 45 and 55%, targets D and F had 30% and 70% GC, respectively. In the revised manuscript we have added a statement with this information on line 373 and indicated the %GC content of the DNA targets in Supplementary Table S1. We did not observe differences in substrate cleavage that correlated with GC content within the scope of our study. Although we tested multiple targets with a range of GC content, we have not assessed target sequence space sufficient to be able to make general conclusions about base or GC content bias.

Conceivably, the question of base-specific or GC content bias could be addressed with the development of a new high-throughput or high multiplex cleavage assay. This could be an interesting future direction to take this work.

5. Figure 6B, the initial fluorescence values (0 min) are inconsistent, both the 5nt and 25nt reporters, especially in the LbaCas12a. Moreover, the 25nt ssDNA reporter showed a higher initial value, is that due to the high background signal?

We thank the reviewer for their keen observation. There is a small amount of lag time between mixing the sample to start the reaction and when the instrument takes its first fluorescent reading. When the reaction is fast, such as with LbaCas12a reactions or AspCas12a with Target W, the initial value isn't zero because the reaction is underway before the "time 0" reading is taken. We have added additional text to the methods section on lines 228-229 to clarify the lag time and its effect on data collection.

REVIEWERS' COMMENTS:

Reviewer #1 (Remarks to the Author):

The author adequately addressed all reservations regarding the manuscript. My opinion is that the manuscript is acceptable for publication.

Reviewer #2 (Remarks to the Author):

Fuchs et al. have fully addressed the comments and have provided additional data exploring the target-DNA mismatch tolerance of the characterized nucleases. This paper is a neat addition to the existing Cas12a literature, and I fully support the publication of this manuscript.

Minor issues:

Figure S2: The inflection points of apo and RNP for Yme and Cme appear to not correspond to the data shown in Fig. 2A. The labels for apo and RNP might have been confused in the nanoDSF traces (for both enzymes, apo unfolds at higher temperatures than the RNP).

line 104: It might be worthwhile to mention that the employed AsCas12a "Ultra" (IDT) is an engineered variant of AsCas12a (M537R/F870L); for reference see <https://www.nature.com/articles/s41467-021-24017-8>

Reviewer #3 (Remarks to the Author):

The author's supplementary experiments and explanations have addressed my concerns.

REVIEWERS' COMMENTS:

Reviewer #1 (Remarks to the Author):

The author adequately addressed all reservations regarding the manuscript. My opinion is that the manuscript is acceptable for publication.

Thank you for taking the time to review our paper and for your helpful feedback throughout the process.

Reviewer #2 (Remarks to the Author):

Fuchs et al. have fully addressed the comments and have provided additional data exploring the target-DNA mismatch tolerance of the characterized nucleases. This paper is a neat addition to the existing Cas12a literature, and I fully support the publication of this manuscript.

Thank you for your thoughtful reviews of our manuscript and for your feedback which has strengthened our paper.

Minor issues:

Figure S2: The inflection points of apo and RNP for Yme and Cme appear to not correspond to the data shown in Fig.2A. The labels for apo and RNP might have been confused in the nanoDSF traces (for both enzymes, apo unfolds at higher temperatures than the RNP).

Thank you for bringing this to our attention, the traces were mislabeled in the version that last sent for review. We have fixed the issue in the revised version.

line 104: It might be worthwhile to mention that the employed AsCas12a "Ultra" (IDT) is an engineered variant of AsCas12a (M537R/F870L); for reference see <https://www.nature.com/articles/s41467-021-24017-8>

We have added a mention of this to the methods section and added the suggested reference.

Reviewer #3 (Remarks to the Author):

The author's supplementary experiments and explanations have addressed my concerns.

We thank the reviewer for their reviews of our manuscript and greatly appreciate their feedback.